# Arctic speleothems reveal nearly permafrost-free Northern Hemisphere in the late Miocene

Anton Vaks [1], Andrew Mason[2], Sebastian F. M. Breitenbach [3] ✉, Alena Giesche [4], Alexander Osinzev[5], Irina Adrian[6], Aleksandr Kononov [7,8], Stuart Umbo [3], Franziska A. Lechleitner [9], Marcelo Rosensaft[10] & Gideon M. Henderson[2]

Arctic warming is happening at nearly four times the global average rate. Long-term trends of permafrost dynamics cannot be estimated directly from monitoring of present-day thaw processes, requiring paleoclimate-proxy information. Here we use cave carbonates (speleothems) from a northern Siberian cave to determine when the Northern Hemisphere was mostly permafrost-free. At present, thick continuous permafrost in this region prevents speleothem growth. In a series of partially eroded caves, speleothems grew during the late Tortonian stage ($8.68 \pm 0.09$ Ma), a time when the geographic position of this site was already similar to today. Paleotemperatures reconstructed from speleothems show that mean annual air temperatures (MAAT) in the region were $+6.6°C$ to $+11.1°C$, when contemporary global MAAT were ~$4.5°C$ higher than modern. Our findings provide direct evidence that warming to Tortonian-like temperatures would leave most of the Northern Hemisphere permafrost-free. This may release up to ~130 petagrams of carbon, enhancing further warming.

About 15% of the Earth's continental surface is underlain by permanently frozen ground (permafrost)[1]. Continued release of anthropogenic greenhouse gases may push atmospheric $CO_2$ and $CH_4$ concentrations to levels at which global temperatures are 2–5°C warmer than pre-industrial levels[2,3] (i.e., higher than the 1.5–2°C above pre-industrial levels cited in international climate agreements[4]). Such level of warming is similar to the one last experienced on Earth in the late Miocene[5]. Under this climate change scenario, thawing of 34–74% of permafrost is expected to occur, mostly in continental Northern Hemisphere regions[6], where about 1672 petagrams of organic carbon (PgC) (about twice as much as in the atmosphere) are stored in the frozen soils[7]. An increase of the mean annual ground temperature (MAGT) above 0°C can initiate decomposition of this carbon with accompanied release of greenhouse gases into the atmosphere, creating a positive feedback in the climate system[7]. Considerable uncertainty still exists, however, about the degree of global warming required to completely thaw the near-surface permafrost presently located in the coldest Northern Hemisphere regions such as northern Siberia, especially considering that Arctic warming is happening at nearly four times the global average rate[8].

Here we use U-Pb dating of vadose speleothems (stalagmites, stalactites, flowstones and cave pool folia) from relict caves on the

[1]Geochemistry and Environmental Geology Division, Geological Survey of Israel, Jerusalem, Israel. [2]Department of Earth Sciences, Oxford University, Oxford, UK. [3]Department of Geography and Environmental Sciences, Northumbria University, Newcastle-upon-Tyne, UK. [4]U.S. Geological Survey, Alaska Science Center, Anchorage, AK, USA. [5]Speleoclub Arabika, Irkutsk, Russia. [6]Lena Delta Wildlife Reserve, Tiksi, Sakha Republic, Russia. [7]Irkutsk National Research Technical University, Irkutsk, Russia. [8]Institute of the Earth's Crust, Russian Academy of Sciences, Siberian Branch, Irkutsk, Russia. [9]Department of Chemistry, Biochemistry and Pharmaceutical Sciences & Oeschger Centre for Climate Change Research, Bern, Switzerland. [10]Geological Mapping Division, Geological Survey of Israel, Jerusalem, Israel. ✉e-mail: sebastian.breitenbach@northumbria.ac.uk

Taba-Ba'astakh cliffs near the Lena delta close to the Arctic Ocean (72°15′N, 126°56′E, Fig. 1) to assess when northern Siberian regions were free of permafrost in the past. Vadose speleothems found in today's permafrost regions attest to warmer periods when continuous permafrost was absent and cave temperatures were above freezing, allowing water seepage into caves[9,10]. Dating such speleothems allows reconstruction of periods of permafrost absence in one of the coldest parts of Siberia, and enables us to understand other key aspects of the past environment in this region, including vegetation dynamics and carbon emissions.

The Taba-Ba'astakh cliffs are located in the Lena Delta Nature Reserve, 97 km north-west of the town of Tiksi that is located on the Arctic coast. The Taba-Ba'astakh site is just upstream of the Lena River delta (Fig. 1, see Methods), >500 km north of the Arctic Circle. The

region is characterized by a tundra climate, with average temperatures ranging between −31.6 °C in January and +9.3 °C in July (mean annual air temperatures (MAAT) of −12.3 °C, Fig. 1)[11], by annual precipitation of ~300 mm[12], by continuous permafrost with thicknesses of 300−500 m, and mean annual ground temperatures (MAGT) of −11 °C to −9 °C[13,14] (see also Methods). The Taba-Ba'astakh cliffs rise 130−140 m above the Lena River banks and are composed of carbonate rocks of Carboniferous age[15]. The caves are found at elevations between 70 and 120 m above the riverbank and are filled with ice, starting a few meters from the cave entrance and preventing access into the cave interiors (see Methods). However, many relict caves with speleothems have been exposed by erosion and are found along the cliffs. Speleothems can be found in situ on the cliff walls, but also as fragments transported downslope onto the riverbank (see Methods).

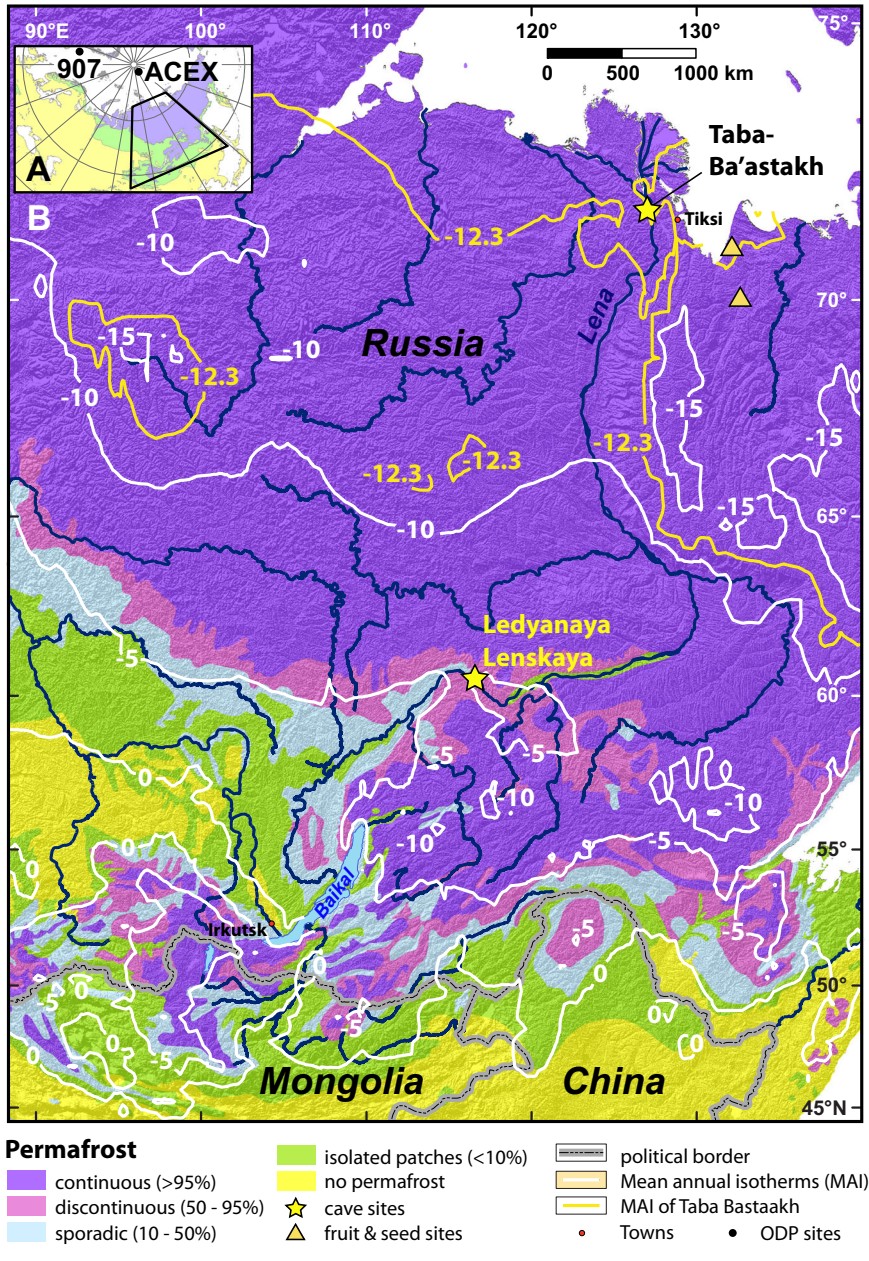

**Fig. 1 | Permafrost map of Siberia[30] with the study sites. A** Map of northern Eurasia with the Arctic Ocean Drilling Program sites 907 and ACEX shown by black circles and the study region as black polygon. **B** Study area enlarged with yellow stars marking the Taba-Ba'astakh site in the north and Ledyanaya Lenskaya Cave (at the present-day boundary of continuous permafrost) in the south[10,16]. Permafrost types are shown by colors as explained in the legend. White lines mark Mean Annual Isotherms (MAI = MAAT = Mean Annual Air Temperature) for every five degrees[11], and the present-day MAAT of the Taba-Ba'astakh site (−12.3 °C) is marked by a yellow isotherm. Miocene fruit and seed flora study sites are shown by yellow triangles[28].

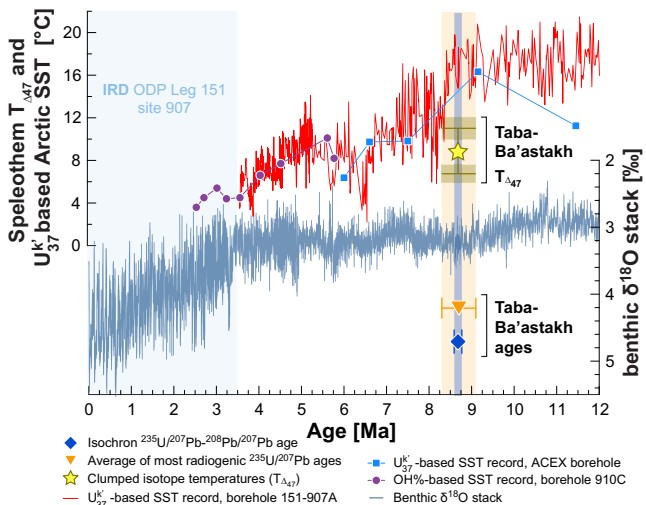

**Fig. 2 | Comparison of speleothem deposition period in Taba-Ba'astakh to other climate records.** The timing of speleothem deposition in Taba-Ba'astakh is shown by the isochron age (blue diamond) with its 2 S.D. uncertainty (blue vertical bold line), and the average age of the 19 most radiogenic samples (yellow triangle) with its 2 S.D. uncertainty (light-yellow shading). The periods of speleothem growth are compared with loess smoothed marine benthic stack $\delta^{18}O$ ($\alpha = 1$)[41] (greyish-blue line), $U^k_{37}$ sea surface temperature (SST) reconstructions from Ocean drilling programme (ODP) site 151–907 (western Arctic, red line, after correcting for minor plate motion, where data show the average of two different calibration reconstructions)[18], OH-GDGT%-SST from ODP site 910 C (purple line and circles)[42] and the ACEX borehole $U^k_{37}$ SST reconstructions (blue line and rectangles)[29]. The Taba-Ba'astakh speleothems' clumped isotopes temperatures range is marked by yellow star with olive error bars (95% confidence ranges). The light-blue shading on the left shows the period when ice-rafted debris appears in ODP-151-907 borehole[43].

## Results

### U-Pb chronology

We analyzed 66 sub-samples from 14 speleothems following the methods described in refs. 16, 17 (see Methods, Dataset-1), to obtain a $^{235}U/^{207}Pb-^{208}Pb/^{207}Pb$ isochron age of $8.68 \pm 0.09$ Ma for the whole dataset, and a mean model $^{235}U/^{207}Pb$ age of $8.7 \pm 0.4$ Ma for 19 analyses of the three most radiogenic samples (see Methods, Dataset-1 and Fig. 2). The total growth interval represented by the samples is ambiguous as many of them are unradiogenic, yielding imprecise model $^{235}U-^{207}Pb$ ages. However, based on the most radiogenic material, a growth interval of no more than a few hundreds of thousands of years around the isochron age is most likely. This time corresponds to the middle-later part the Tortonian stage (11.63–7.25 Ma), late Miocene, with speleothem deposition taking place immediately before the first major cooling event of Arctic Ocean sea surface temperatures (SST) at 8.3–8.1 Ma[18] (Fig. 2).

### Comparison of paleotemperature records

The presence of multiple middle-late Tortonian speleothems at the Taba-Ba'astakh site shows that at $8.68 \pm 0.09$ Ma this present-day tundra region experienced a more temperate climate, with MAGT above 0 °C and with permafrost-free conditions. The geographical position of eastern Siberia did not change significantly over the last 80 Ma[19], suggesting that plate-movement-related latitude changes did not affect climate in the region. Thus, warmer climate conditions in this context are the result of global climate change compared to present. The minimum MAAT range that leads to discontinuous permafrost and enables water penetration into the caves is 0 °C to −6 °C[20], however, clumped isotope analyses performed on four Taba-Ba'astakh speleothems show paleotemperatures (MAGT) of +6.6 °C to +11.1 °C[21]. During the Tortonian some tree growth extended to 80°N[22] and in

forested regions with MAAT ≥ +5 °C, the MAGT are close to MAAT[23]. The temperatures between +6.6 °C and +11.1 °C[21] range between present-day MAAT in Stockholm[24] and London[25], respectively, and it follows that the climate was temperate. If permafrost was absent at this site and further south, this indicates that most of the Siberian landmass and likely similar regions in the Northern Hemisphere were permafrost-free when speleothems formed at Taba-Ba'astakh.

The observation of middle-late Tortonian permafrost-free Siberia adds important details to the context of differing paleotemperature reconstructions for this time. During the Seravallian stage (13.82–11.63 Ma), preceding the Tortonian growth of Taba-Ba'astakh speleothems, temperate mixed forests covered large areas of subarctic Asia between 61°N and 77°N[26]. Although climate cooled during the Tortonian[5], palynological evidence from continental fluvial/lacustrine sediments suggests that coniferous forests reached as far as 70°N[26], with some tree growth extending to 80°N, i.e., 10° further north than today[22]. Vegetation models for this time interval indicate that global MAAT may have been up to 4.5 °C higher than present[5,27], while warming of the Arctic was amplified by more than 9 °C[26]. Clumped isotope paleotemperatures from Taba-Ba'astakh speleothems show MAAT of +6.6 °C to +11.1 °C[21], indicating a warming of 18.9 °C to 23.4 °C above present, considering the difference between the Tortonian MAAT and present day MAAT of −12.3 °C. This level of warming is similar to values estimated for Tortonian by reconstructions based on fruit and seed remains from sediments from the Omoloy River (70°N, 133°E, ca. 350 km SE of Taba-Ba'astakh), that indicate MAAT between +7 °C and +16 °C, with a mean temperature of coldest month of −4 °C[28]. Flora from the sediments from nearby Temmirdekh-Khaja (71°N, 132°E, ca. 230 km E from Taba-Ba'astakh) suggests a MAAT between +9.3 °C and +10.8 °C, with a coldest month mean temperature ranging from −3 °C to +1 °C, and warmest month mean temperature of about 22 °C[28]. Summer SST estimates from ODP site 907 in the western Arctic ocean (69°15'N, 012°42'W) range between +13 °C and +20 °C in the early-middle Tortonian, before decreasing to ~+6 °C to +13 °C at 8.3–8.1 Ma[18,29]. SSTs from IODP site 302-ACEX in the central Arctic (Lomonosov Ridge, 87°54'N, 138°39'E) were around +16 °C in the middle Tortonian (~9.1 Ma) before decreasing to ~+10 °C in the late Tortonian (~7.5 Ma)[29]. These lines of evidence imply a temperate climate and an ice-free Arctic Ocean (at least during summer) in the Tortonian[29] (Fig. 2). Under such climate conditions, only brief, if any, ground freezing in mid-winter would occur, not hampering speleothem growth that was enhanced by high Miocene mean-annual precipitation of 800–900 mm[28].

### Reconstruction of permafrost and carbon emissions

To assess the potential loss of carbon stored in Northern Hemisphere frozen soils under the range of temperatures consistent with the absence of permafrost at Taba-Ba'astakh, we compared permafrost extent[30] and modelled soil organic carbon stocks in the top 3 m[31] of northern permafrost regions with modern patterns of Siberian land temperature (2 m Global Historical Climatology Network (GHCN) land temperature from 1991 to 2020[11], Fig. 3, Dataset-2, Methods). Considering today's MAAT of −12.3 °C at Taba-Ba'astakh, we chose an increase of 18.9 °C above present in polar regions of the Northern Hemisphere, to MAAT of +6.6 °C obtained from clumped isotope measurement of Taba-Ba'astakh speleothems[21]. This is our conservative estimate of warming during the Tortonian (Fig. 3A, B, Dataset-2, Methods).

Relating our paleoclimate data from the Tortonian to future warming scenarios in Siberia suggests that most to all of the surficial permafrost would thaw with 18.9 °C of warming, with only 1% of present-day permafrost areas remaining as discontinuous permafrost (the purple regions in northern Greenland and Ellesmere Island in Fig. 3A). Model scenarios of 20 °C warming or higher (relative to present)[21] show that no surficial permafrost would remain in the

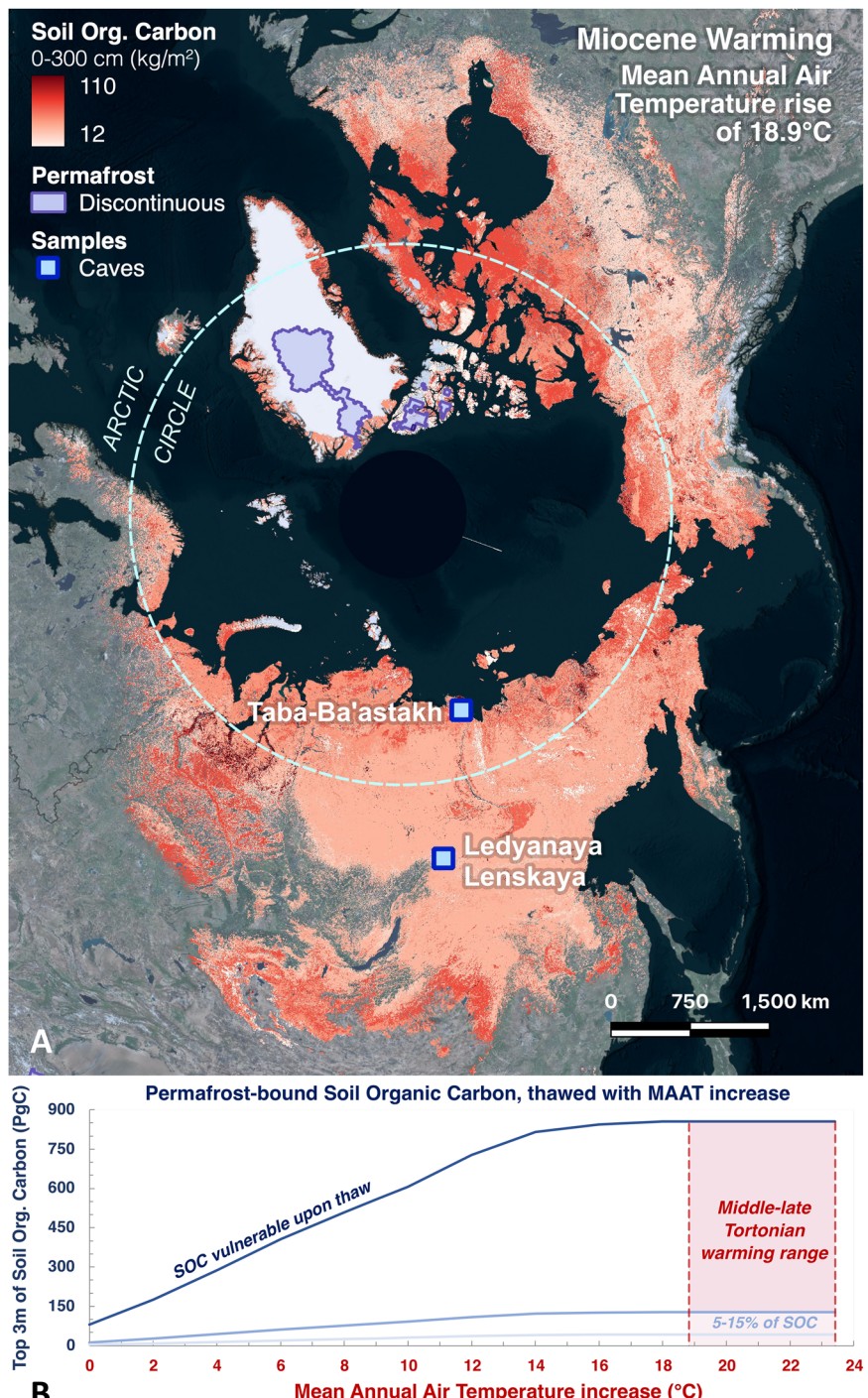

**Fig. 3 | The permafrost area and its boundaries modelled for 18.9 °C of warming relative to present day, representative of the middle-late Tortonian (8.7 ± 0.4 Ma B.P.). A** Areas sustaining discontinuous permafrost are marked by light purple (northern Greenland and Ellesmere Island). This is compared to present day permafrost extent (area marked by red, with darker reds representing higher amounts of soil organic carbon contained in the top 3 meters[31]). In the case of 23.4 °C warming (not mapped) these small areas of remaining discontinuous permafrost would disappear too, leaving the entire Northern Hemisphere permafrost-free. The caves with speleothem-based paleoclimate studies cited in the current research are shown by the blue rectangles. **B** The graph depicts soil organic carbon (SOC) vulnerable to thaw at a given mean annual air temperature (MAAT) increase (dark blue line) and the likely proportion[33,34] of SOC released as carbon to the atmosphere within decades to centuries (light blue line), highlighting the Tortonian MAAT warming range (red dashed lines).

Northern Hemisphere (see Methods). If all near-surface permafrost existing today in the Northern Hemisphere would thaw, an estimated ~855.42 PgC from the top 3 m of soil would become vulnerable for mobilization (Fig. 3B). This estimate is an underestimate of the total vulnerable carbon, considering that additional organic carbon seated deeper than 3 m might be mobilized, although the relative contribution of this deeper carbon stock to the total release remains poorly constrained. While some of the stored soil carbon would remain sequestered in the thawed landscape, a portion would be emitted as greenhouse gases (e.g., carbon dioxide and methane). Estimating

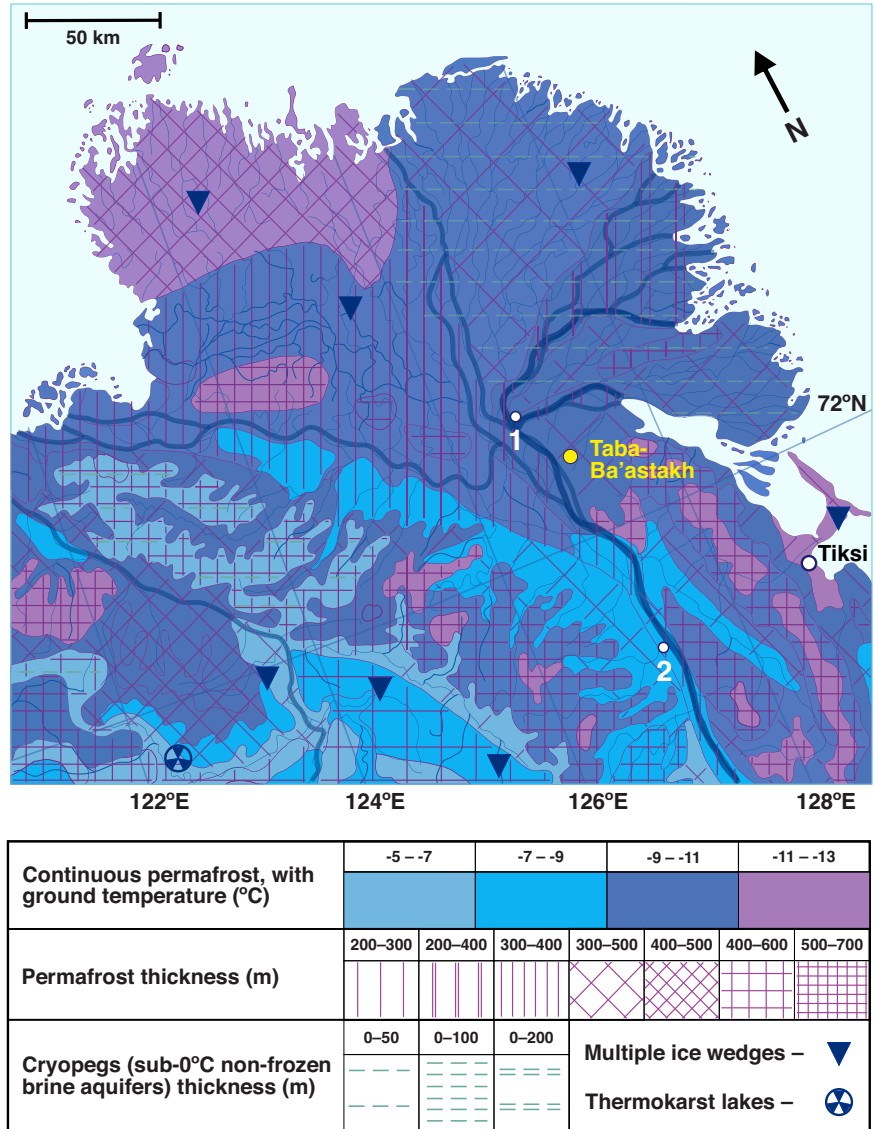

**Fig. 4 | Permafrost map of the Lena River delta[13,14] area shown with the locations of the Taba-Ba'astakh research site (yellow circle) and the town of Tiksi (white circle).** Points 1 and 2 show the northern and southern edges (respectively) of the section of the Lena River where the search for caves and speleothems took place during the expedition. The legend shows types of continuous permafrost with its ground temperatures, permafrost thickness, cryopegs (brine sub-0 °C aquifers) thickness and other geological features. Map reproduced with kind permission from the Geocryology Department of the Geology Faculty at Lomonosov Moscow State University.

permafrost carbon release and uptake is uncertain because it depends on heterogeneous landscape factors such as ice content, abundance and distribution of lakes, wetlands, and vegetation composition that shift as the landscape evolves[32]. Schuur et al. [33,34] and references within suggest that 5–15% of this surficial soil organic carbon would be vulnerable to be released as greenhouse gases within decades in the short-term, while longer-term estimates for soil carbon release are poorly constrained. Therefore, most conservatively 42.77–128.31 PgC will be released into the atmosphere for uniform warming of 18.9 °C and higher across the Northern Hemisphere permafrost region (Fig. 3B).

## Discussion

The causes of globally warmer conditions in the Tortonian are complex and differ from present-day warming driven by increased $CO_2$ levels. Assessments of past atmospheric $CO_2$ concentrations during the late Miocene suggest values in the range from ~300 to ~600 ppm[5,22,35], with hotter climates partly related to global geographic differences. For example, the Panama seaway between the Atlantic and the Pacific was

still at least partially open[22], the Bering Strait was closed[36], and the connection between the Arctic and Atlantic oceans was likely more restricted, resulting in a less saline Arctic Ocean with limited influx of warm and salty Atlantic water[37]. Whilst such boundary conditions differed to those of the present-day, a permafrost-free Tortonian Siberia still represents a scenario with ~4.5 °C warming of global MAAT, closest to the most extreme warming scenarios proposed by the IPCC[2,3]. This adds to findings by ref. 38, who assessed middle Pliocene warm period (3.264–3.025 Ma) temperature data where global annual mean surface air temperature was 3.2 °C higher than preindustrial and 7.2 °C higher in Arctic regions, and demonstrated a 93 ± 3% reduction in permafrost extent. The absence of near-surface permafrost at 72°N under such conditions implies that future warming of similar magnitude is likely to cause permafrost thaw of similar proportion in almost all continental areas of the Northern Hemisphere.

Previous work on speleothems from southern Siberia showed that the southern boundary of continuous permafrost (Ledyanaya Lenskaya Cave; 60°22'N-116°56'E; Figs. 1B and 3A, B) reached its present-day

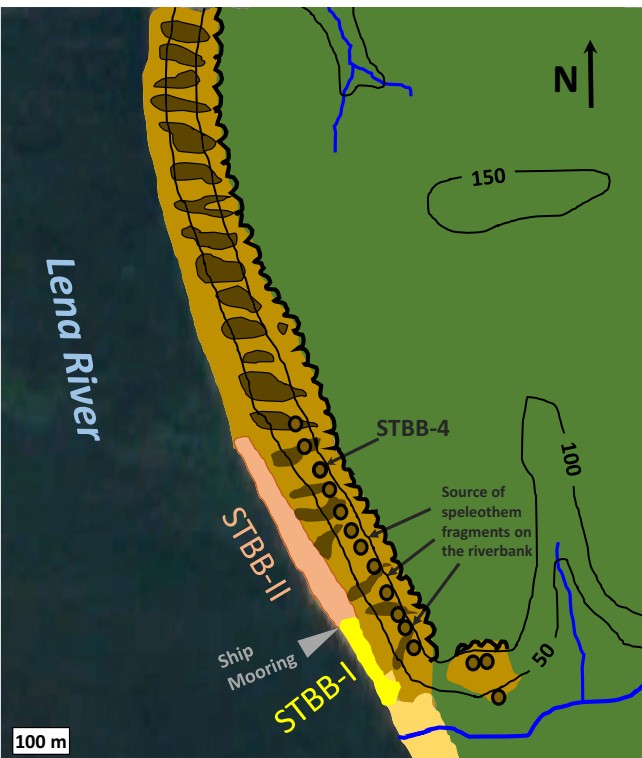

**Fig. 5 | Map of the Taba-Ba'astakh cliffs and Lena River bank with locations of sites where the speleothems were collected.** The moss-grass-shrub tundra is marked by green, the Lena River is in the black area, and thin blue lines in the green area represent small inlet streams. Black contour lines of 50, 100 and 150 m show the elevation above sea level. The cliffs are shown in brown-beige, the sites with in-situ speleothems are shown by circles, including the STBB-4 site. Two sections of the riverbank where speleothem fragments were collected, STBB-I and STBB-II, are shown by yellow and pink, respectively, with the ship mooring site in between.

southward extent ~400 ka ago[10,16]. This happened when year-round sea ice cover was established in the Arctic Ocean[16]. Continuous sea-ice cover limited the transport of heat and moisture from the open Arctic ocean surface into the Siberian landmass, reducing winter temperatures and snow cover, thus lowering ground temperatures and fostering the formation and stability of permafrost[16]. These studies suggested that the disappearance of Arctic summer sea ice and associated northward retreat of the continuous permafrost boundary required a global MAAT ~ 1.5 °C higher than preindustrial levels[10,16]. Our present study extends that work to show that mean global surface temperatures of ~4.5 °C above preindustrial level could result in the elimination of near-surface permafrost in most of the terrestrial Arctic, rendering almost the entire Northern Hemisphere landmass permafrost-free.

## Methods
### Research site description
The Taba-Ba'astakh Cliffs are located in northern part of Kharaulah Ridge (northern part of Verkhoyansk Mountains), 97 km north-west of the town of Tiksi, Bulunsky District, Sakha Republic, Russian Federation (Fig. 1, Fig. 4). The area is characterized by a typical tundra climate, with average temperatures varying between −31.6 °C in January and +9.3 °C in July (MAAT of −12.3 °C)[11] and annual precipitation of ~300 mm[12]. This region is underlain by continuous permafrost with a thickness of 300–500 m[13,14] and with MAGT of −9 °C to −11 °C[13,14] (Fig. 4). The eastern Siberian geographic position was relatively stable during the last 80 My, therefore this region is very suitable for long-term paleoclimate studies[19]. The expedition to Taba-Ba'astakh and other carbonate outcrops near the Lena River mouth was undertaken

aboard the ship "Orlan" belonging to the Ust-Lena Nature reserve and based in Tiksi Port. The expedition took place between 05 and 15 August 2014, and it examined cliffs and riverbanks along a ~ 90 km route from the Stolb (Pillar) Island in the Lena River delta (72°23'42.53"N, 126°39'34.27"E, point 1, Fig. 4) to the Khatistakh River mouth (71°36'39.95"N, 127°14'27.84"E, point 2, Fig. 4). Speleothems were found only on the Taba-Ba'astakh Cliffs (72°16'09.7"N, 126°56'15.7"E, Figs. 4, 5 and 6). These cliffs rise 140 m high above the Lena River banks and are composed of carbonate rocks of Carboniferous age[15] (Fig. 6A, B). Speleothems on relict walls of eroded caves are found at elevations of 70 to 120 m above the riverbank (Fig. 6). A few hundred meters to the south of these exposed speleothem sites, an existing karst system forms a group of cave entrances at a similar elevation as the eroded caves (Fig. 6C, D). However, ice infilling of these caves makes exploration of passages deeper than 10–15 m impossible (Fig. 6E).

The geographical position of eastern Siberia did not change significantly over the last 80 Ma[19]. In terms of vertical movements, during the middle Miocene, the region underwent compression that contributed to the uplift of the mountain ridges we see today[39]. The speleothems on the cliffs are flowstone, stalactites, stalagmites, and folia (Fig. 6) composed of calcite. Speleothem thickness varies from several cm to tens of cm. Speleothem samples from Taba-Ba'astakh were divided into three groups according to the location in which they were found: STBB-I, STBB-II and STBB-4 (Figs. 5 and 6). Sample groups STBB-I and STBB-II include fragments that fell to the riverbank from the currently eroded caves on the cliff. The STBB-I speleothems were fragments that fell from the southern part of the Taba-Ba'astakh Cliffs and were collected south of the expedition ship-mooring location (72°16'0.44"N, 126°56'23.71"E, Figs. 5 and 6). The STBB-II speleothems were collected to the north of the mooring and fell from the northern part of Taba-Ba'astakh Cliffs (Figs. 5 and 6). The third group includes STBB-4 speleothems collected from the site "Cave-4" (72°16'09.0"N, 126°56'15.7"E) 70–120 m above the riverbank were STBB-II speleothems were collected (Figs. 5 and 6). In total, 14 speleothems were selected for U-Pb dating, based on preservation conditions. Selection was based on visual inspection and only samples with light transparent clean calcite with little or no weathering-related features were considered for dating purposes.

### Speleothem mineralogy and petrography
The speleothems were sectioned using a diamond saw to expose their petrography. Speleothem mineralogy was examined using a Rigaku SmartLab II X-ray diffractometer, at Northumbria University, with a Copper K-α radiation source at 1.5406 Å with scan in Bragg Brentano mode. The results show that all speleothems are composed of calcite. Visual inspection shows that speleothems consist of multiple brown, beige or grey calcite horizons and the speleothem calcite is usually dense, with minimal porosity (Fig. 7A–P). Microscopic inspection shows that all fourteen speleothems are characterized by calcite with a typical columnar crystal structure (Fig. 7O, P). Some minor primary porosity was found in flowstone STBB-II−9, while flowstone STBB-II−1 shows whitish powdery carbonate on its margins, indicative of weathering dissolution processes that affected this sample from the outside (Fig. 7Q–T). Therefore, sampling for chronology in the latter two speleothems was performed from sections that were not affected by these weathering signs.

### U-Pb chronology
U-Pb chronology of speleothems was performed using the methods, half-lives and age-calculation assumptions described in refs. 16, 17. Sixty-six sub-samples were analysed to determine $^{238}U$-$^{206}Pb$, $^{208}Pb/^{206}Pb$, $^{235}U/^{207}Pb$, $^{208}Pb/^{207}Pb$ and $^{207}Pb/^{206}Pb$ ratios (Excel Dataset-1). Both the $^{238}U$-$^{206}Pb$ and $^{235}U$-$^{207}Pb$ decay systems are considered here. The data for the former system are of higher precision and have a smaller correction

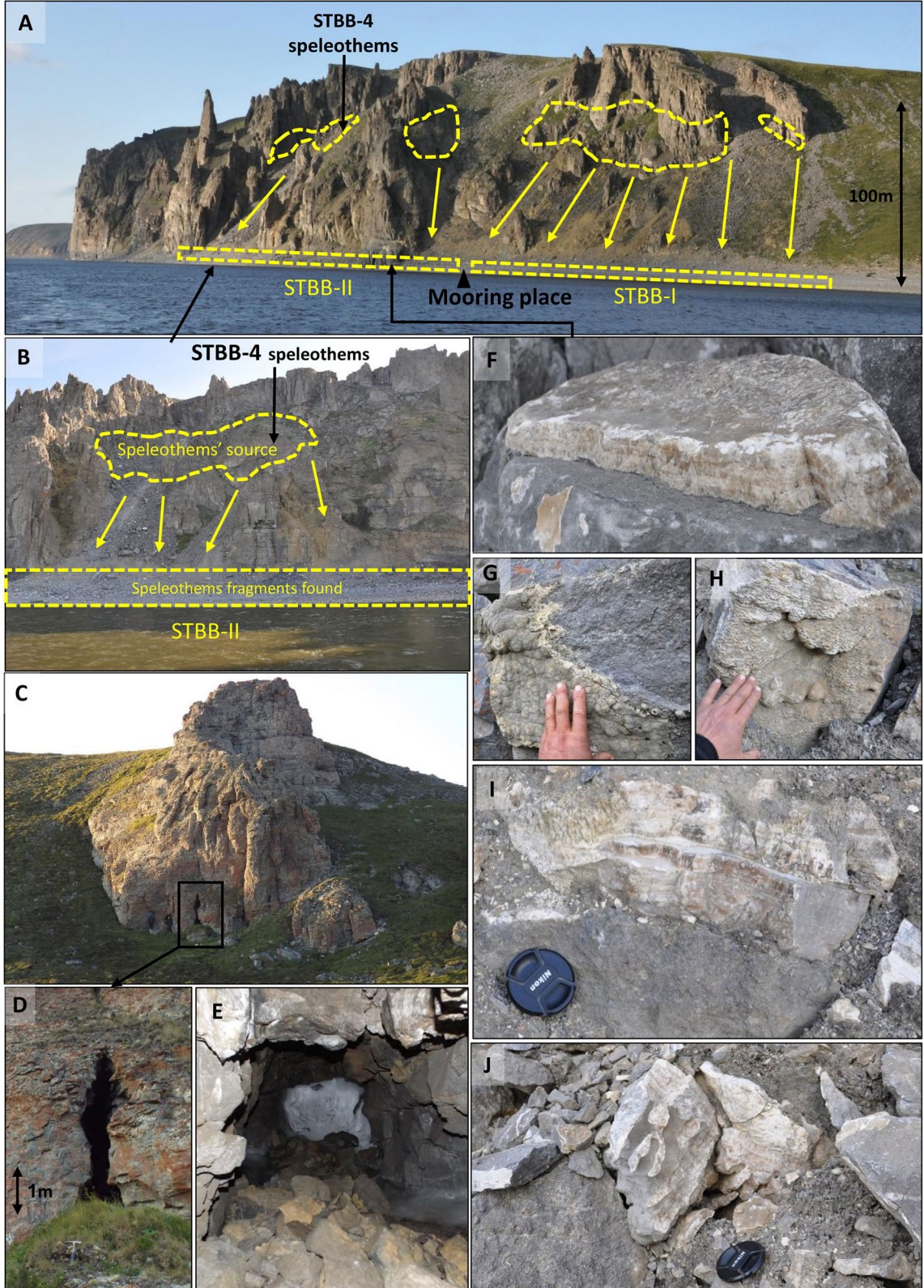

**Fig. 6 | Overview of sampling sites within the Taba-Ba'astakh cliffs. A** Picture of the Taba-Ba'astakh Cliffs with locations of areas on the cliff where speleothems were located. The cliffs' height is 120–140 m, and the speleothem fragments that fell from the cliff were collected on the riverbanks STBB-I and STBB-II, with the ship mooring point in between. **B** Frontal view of speleothem sampling site STBB-4 and STBB-II. **C, D** Karst caves located on the southern edge of the Taba-Ba'astakh cliffs (about 50–60 m below the top of the cliff). The ice filling in caves begins 10–15 m from the entrance (**E**). One of the rock fragments with a flowstone attached (STBB-II area) is shown in (**F**). Speleothems on the cliff at the STBB-4 site include mamillaries (**G**), flowstone (**H, I**), stalactites (**J**) (Photos by Osinzev, A. and Vaks, A.).

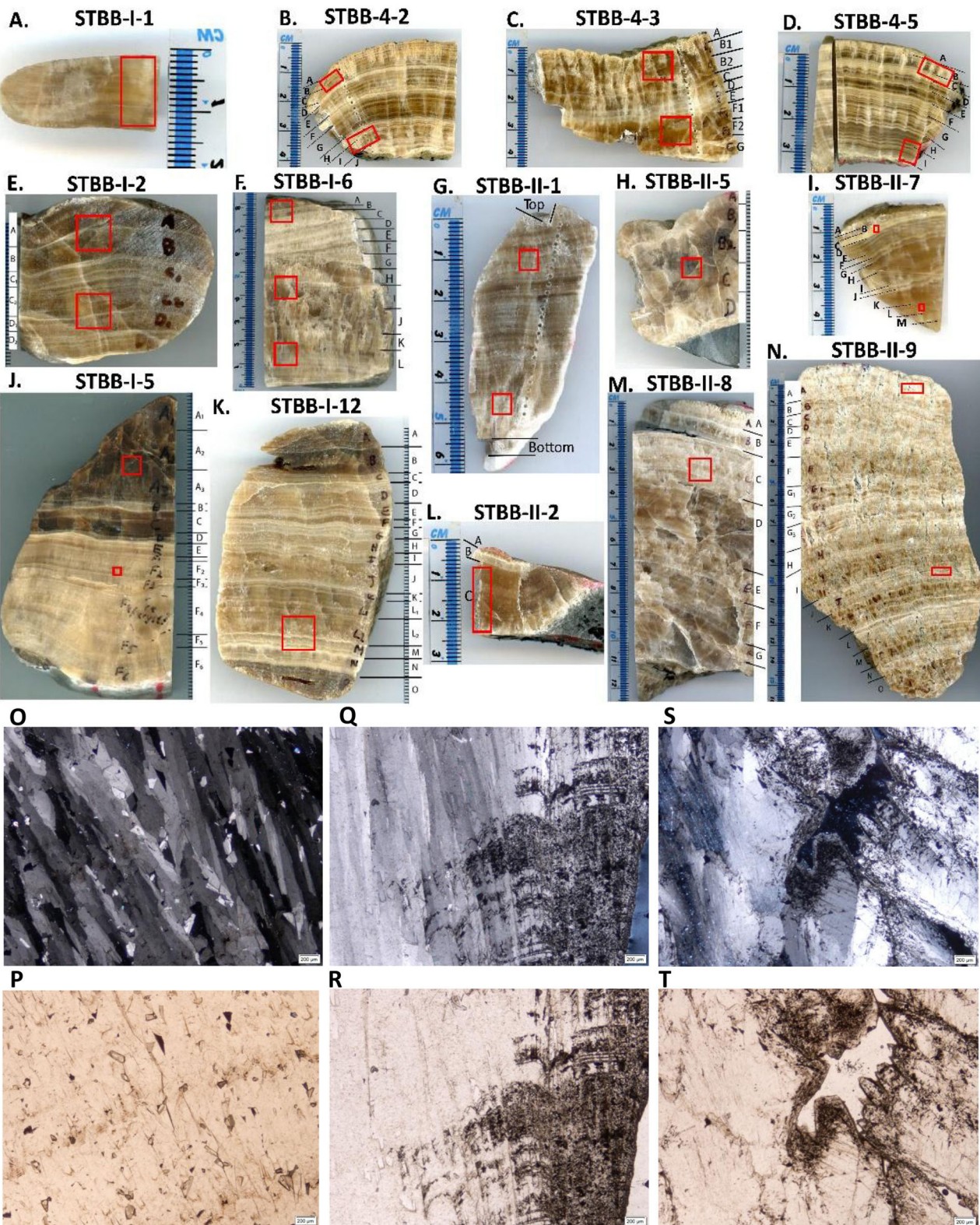

**Fig. 7 | Sections of speleothems used in the study with marked growth layers.** Flowstone STBB-I-1 (**A**); Flowstone STBB-4-2 (**B**); Folia STBB-4-3 (**C**); Flowstone STBB-4-5 (**D**); Flowstones STBB-I-2 (**E**); STBB-I-6 (**F**); STBB-II-1 (**G**); STBB-II-5 (**H**); STBB-II-7 (**I**); STBB-I-5 (**J**); STBB-I-12 (**K**); STBB-II-2 (**L**); STBB-II-8 (**M**); and STBB-II-9 (**N**). Sampling for X-ray diffraction analysis is marked by red squares. The examples of speleothems' petrography under the polarizing microscope usually showed the typical columnar crystal structure with no weathering signs, like in flowstone STBB-I-12 (O – cross-polar-light (cpl); P – plain-polar-light (ppl)). In two speleothems, STBB-II-1 and STBB-II-9 along the usual petrography, some weathering signs were found. These include micritization near the surface (STBB-II-1, Q – cpl; R – ppl), or porosity (STBB-II-9, S - cpl; T - ppl). While sampling for chronology, these weathered places were avoided.

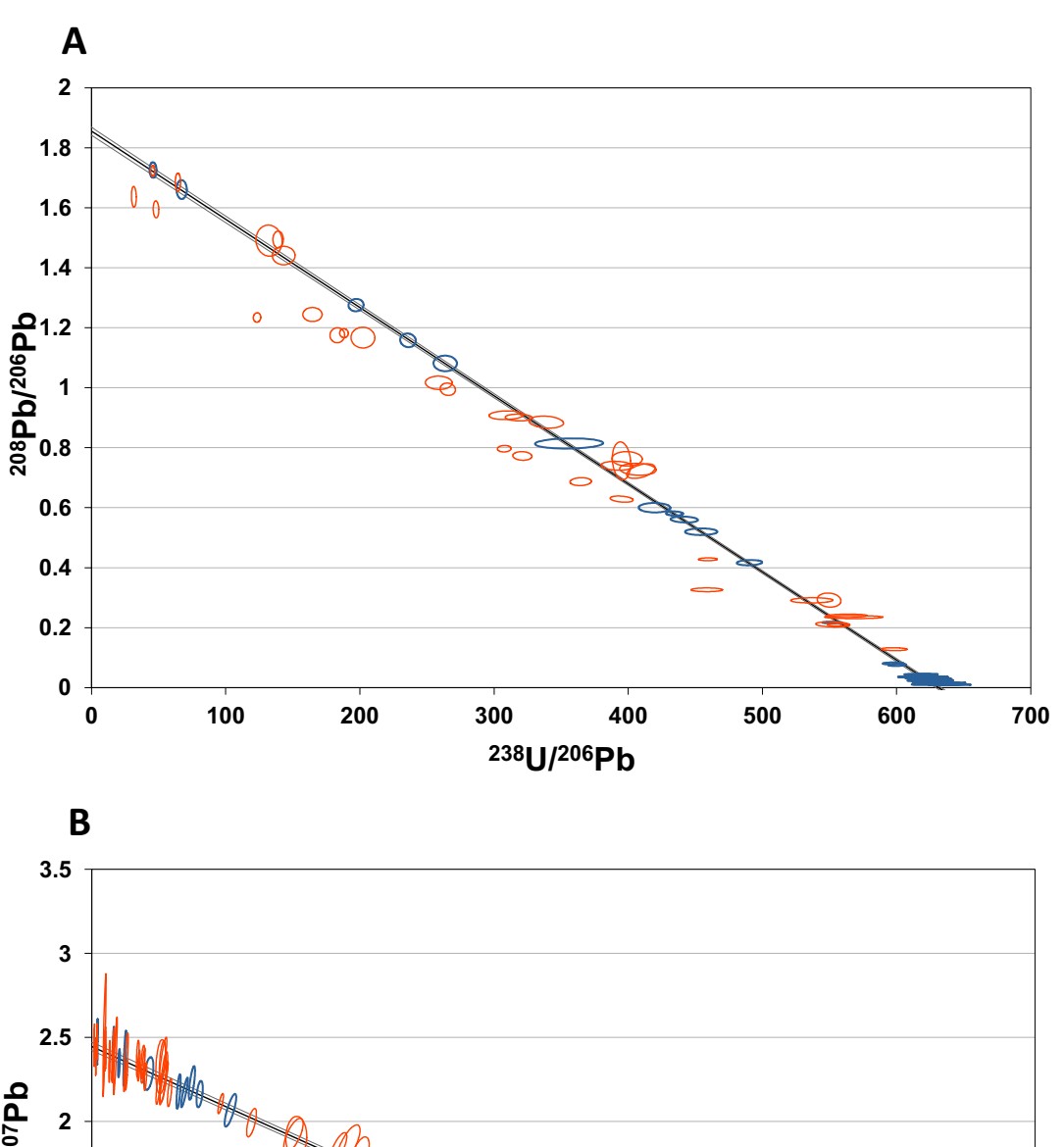

for initial (common) Pb than for the latter system, making $^{238}$U-$^{206}$Pb ages potentially more precise (and potentially more sensitive to intra and inter sample age variation) than the $^{235}$U-$^{207}$Pb ages. However, obtaining accurate absolute $^{238}$U-$^{206}$Pb ages is heavily reliant on knowledge of the initial $^{234}$U/$^{238}$U ratio, variation of which produces age-independent variation in the $^{238}$U/$^{206}$Pb ratio, and, thus, spurious $^{238}$U-$^{206}$Pb ages if not accounted for. In the case of the Taba-Ba'astakh speleothems, no young samples providing independent constraint on the initial $^{234}$U/$^{238}$U ratio

are available. Thus, the strategy here is to use the more precise $^{238}$U-$^{206}$Pb data to help identify/exclude potential age variations within the data set, while focusing on the $^{235}$U-$^{207}$Pb system to obtain accurate absolute ages free of the uncertainty associated with an unconstrained initial $^{234}$U/$^{238}$U ratio.

The measured isotopic ratios of all samples are plotted in $^{238}$U/$^{206}$Pb-$^{208}$Pb/$^{206}$Pb space (Fig. 8A) and $^{235}$U/$^{207}$Pb-$^{208}$Pb/$^{207}$Pb space (Fig. 8B). In Fig. 8 the data can be divided into two populations. All

**Fig. 8 | Isochron diagrams showing all Taba-Ba'astakh chronology data.** In $^{238}U/^{206}Pb$-$^{208}Pb/^{206}Pb$ space (**A**) all data from samples STBB-II-8, STBB-II-2, STBB-4-2, STBB-4-5, STBB-I-1, STBB-II-5 and STBB-II-7 (selected data in blue) form a near-perfect linear array suggesting that these samples are undisturbed, and have a uniform initial $^{234}U/^{238}U$ ratio, uniform age and common Pb composition. Note that the absolute age for these samples cannot be unambiguously determined solely from the $^{238}U$-$^{206}Pb$ system in this instance, because it is not possible to characterise the initial $^{234}U/^{238}U$ ratio without additional information. The data from the remaining samples (red) shows considerable excess scatter attributable to open-system behavior, variation in age, variation in initial $^{234}U/^{238}U$ ratio, or variation in common Pb composition, in some combination. In $^{235}U/^{207}Pb$-$^{208}Pb/^{207}Pb$ space (**B**) all data ellipses but one are located on one isochron line. The data from samples

STBB-II-8, STBB-II-2, STBB-4-2, STBB-4-5, STBB-I-1, STBB-II-5 and STBB-II-7 are in blue, data from other samples in red. As in $^{238}U/^{206}Pb$-$^{208}Pb/^{206}Pb$ isotope space, samples STBB-II-8, STBB-II-2, STBB-4-2, STBB-4-5, STBB-I-1, STBB-II-5 and STBB-II-7 form a near-perfect linear array, again suggesting a common $^{235}U/^{207}Pb$ age of c. 8.8 Ma for these samples. Data from other samples are also mostly consistent with the same isochron and show less scatter in $^{235}U/^{207}Pb$-$^{208}Pb/^{207}Pb$ isotope space than $^{238}U/^{206}Pb$-$^{208}Pb/^{206}Pb$ isotope space suggesting that some of the scatter in the latter is probably attributable to variation in the initial $^{234}U/^{238}U$ ratio, which does not affect the $^{235}U$-$^{207}Pb$ system. A common $^{208}Pb/^{207}Pb$ value of 2.45 ± 0.15 is estimated based on the isochron fit to samples STBB-II-8, STBB-II-2, STBB-4-2, STBB-4-5, STBB-I-1, STBB-II-5 and STBB-II-7.

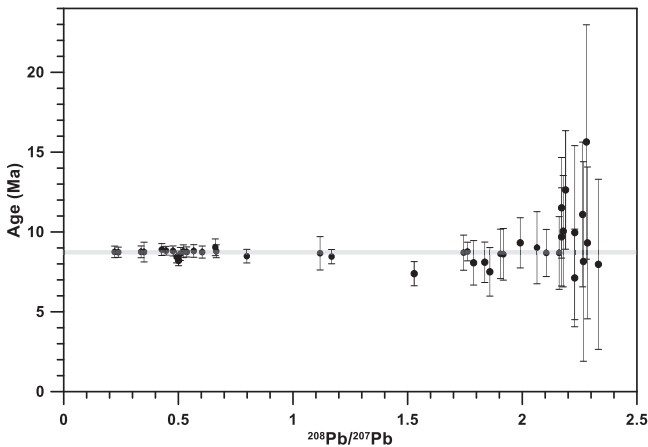

**Fig. 9 | Analytical uncertainty as function of non-radiogenic $^{207}Pb$ in the samples.** The isochron $^{208}Pb/^{207}Pb$-$^{235}U/^{207}Pb$ age of 8.68 ± 0.09 Ma is marked by the grey line. The samples become less radiogenic from left to right, causing the $^{208}Pb/^{207}Pb$ ratio to increase from 0 (left) to 2.5 (right). The age uncertainty (vertical error bars, 2σ) becomes larger in this direction as well.

analyses from samples STBB-II-8, STBB-II-2, STBB-4-2, STBB-4-5, STBB-I-1, STBB-II-5 and STBB-II-7 (data in blue) are mutually co-linear (MSWD = 1.0), suggesting these samples form a closed system in terms of U-Pb isotopes, with a uniform age, similar initial $^{234}U/^{238}U$ ratio, and similar common Pb composition (common $^{208}Pb/^{206}Pb$ = 1.855 ± 0.013, based on the isochron intercept). Analyses from other samples (marked in red) do not fall consistently on the isochron defined by the aforementioned samples, and show significant excess scatter if regressed (MSWD 53). This scatter may result from any of the following factors: open system behaviour, real age variations within and between these samples, variation in the initial $^{234}U/^{238}U$ ratio (hence age-independent variation in the radiogenic $^{238}U/^{206}Pb$ ratio), or variation in the common $^{208}Pb/^{206}Pb$ ratio, in some combination.

Examining the analyses in $^{235}U/^{207}Pb$-$^{208}Pb/^{207}Pb$ space (Fig. 8B), samples STBB-II-8, STBB-II-2, STBB-4-2, STBB-4-5, STBB-I-1, STBB-II-5 and STBB-II-7 (data in blue) are again co-linear (MSWD = 0.88) suggesting a common Pb $^{208}Pb/^{207}Pb$ ratio of 2.447 ± 0.025 and isochron $^{235}U/^{207}Pb$-$^{208}Pb/^{207}Pb$ age of 8.710 + 0.091−0.089 Ma. It is worth noting that the $^{235}U/^{207}Pb$-$^{208}Pb/^{207}Pb$ isochron age is controlled mostly by the two radiogenic samples STBB-II-8 and STBB-II-2. However, the consistent co-linearity of the data from the samples STBB-4-2, STBB-4-5, STBB-I-1, STBB-II-5 and STBB-II-7 with those from samples STBB-II-8 and STBB-II-2 in both $^{235}U/^{207}Pb$-$^{208}Pb/^{207}Pb$ space and $^{238}U/^{206}Pb$-$^{208}Pb/^{206}Pb$ space (where the data are more sensitive to age variations) suggests an age for the samples STBB-4-2, STBB-4-5, STBB-I-1, STBB-II-5 and STBB-II-7, similar to the obtained $^{235}U/^{207}Pb$-$^{208}Pb/^{207}Pb$ isochron age of 8.710 + 0.091−0.089 Ma.

Including the remaining data in the $^{235}U/^{207}Pb$-$^{208}Pb/^{207}Pb$ regression (data in red in Fig. 8) still returns a fairly good isochron fit and an indistinguishable $^{235}U/^{207}Pb$-$^{208}Pb/^{207}Pb$ age (MSWD = 1.2;

8.684 + 0.087−0.085 Ma), with all but one of these additional data falling on the isochron defined by samples STBB-II-8, STBB-II-2, STBB-4-2, STBB-4-5, STBB-I-1, STBB-II-5 and STBB-II-7. This suggests that most of the additional samples have remained a closed system and have a similar (or at least not resolvably different) age, with the scatter in the data in $^{238}U/^{206}Pb$-$^{208}Pb/^{206}Pb$ space being largely attributable to variations in the initial $^{234}U/^{238}U$ ratio (which affects the $^{238}U$-$^{206}Pb$ system but not the $^{235}U$-$^{207}Pb$ system). However, there is a caveat here in that around 50% of the data with excess scatter in $^{238}U/^{206}Pb$-$^{208}Pb/^{206}Pb$ space (i.e., the data in red in Fig. 8) plot close to the non-radiogenic end of the array in $^{235}U/^{207}Pb$-$^{208}Pb/^{207}Pb$ space. This means that their position is controlled mostly by the common Pb $^{208}Pb/^{207}Pb$ ratio, rather than their age – i.e., theoretically there could still be real age variations that are not resolvable in $^{235}U/^{207}Pb$-$^{208}Pb/^{207}Pb$ space.

Using common Pb compositions based on the above isochrons, it is possible to calculate model $^{235}U$-$^{207}Pb$ ages for the more radiogenic analyses, and to solve the $^{238}U$-$^{206}Pb$ decay equation for the initial $^{234}U/^{238}U$ ratio using the corresponding $^{235}U$-$^{207}Pb$ age. In terms of the potential growth interval collectively represented by the Taba-Ba'astakh samples, all but one of the 66 analyses fall on the isochron in $^{208}Pb/^{207}Pb$-$^{235}U/^{207}Pb$ space (Fig. 8) such that the samples are mostly unresolvable from having formed instantaneously at the c. 8.7 Ma isochron age (Fig. 9). However, the resolution with which potential age variations could be detected varies greatly between the samples depending on their radiogenicity (Fig. 9). For many less radiogenic samples, the large model age uncertainty arising from the common Pb correction negates using them in any meaningful way in constraining the growth interval recorded (Fig. 9). The most radiogenic analyses suggest a growth interval no longer than a few hundred-thousand of years, but represents only 3 of 14 speleothems (see below). With only a minority of the samples being adequately datable, little else can be said with confidence about the growth interval.

$^{235}U$-$^{207}Pb$ ages of individual samples are shown in Dataset-1. Nineteen of the most radiogenic analyses (samples STBB-II-8, STBB-II-2, and STBB-I-1) yield the most precise individual $^{235}U$-$^{207}Pb$ ages, clustering around the mean of 8.702 ± 0.378 Ma (2 S. D., n = 19), with the uncertainty of the mean mostly reflecting the precision of the individual model ages, rather than age variation between speleothems. Initial $^{234}U/^{238}U$ activity ratios in samples that have relatively high proportions of radiogenic Pb are also shown in Dataset-1. The most radiogenic speleothems STBB-II-8, STBB-II-2, STBB-I-1 have an average initial $^{234}U/^{238}U$ activity ratio of 5.46 ± 1.19 (2σ). This estimate does not include representation of the analyses that show significant excess scatter in $^{238}U/^{206}Pb$-$^{208}Pb/^{206}Pb$ space (Fig. 8A, data in red) (these analyses are generally too non-radiogenic to obtain precise $^{235}U$-$^{207}Pb$ ages to calculate usable initial $^{234}U/^{238}U$ ratios). However, based on the strong enrichment of initial $^{234}U$ in samples STBB-II-8, STBB-II-2, and STBB-I-1, relatively large variations in initial $^{234}U/^{238}U$ ratio could reasonably be expected within the whole data set (cf. Botovs-kaya cave; refs. 16, [17], and refs. therein), consistent with scatter seen in the $^{238}U/^{206}Pb$-$^{208}Pb/^{206}Pb$ data, but not in the $^{235}U/^{207}Pb$-$^{208}Pb/^{207}Pb$ data.

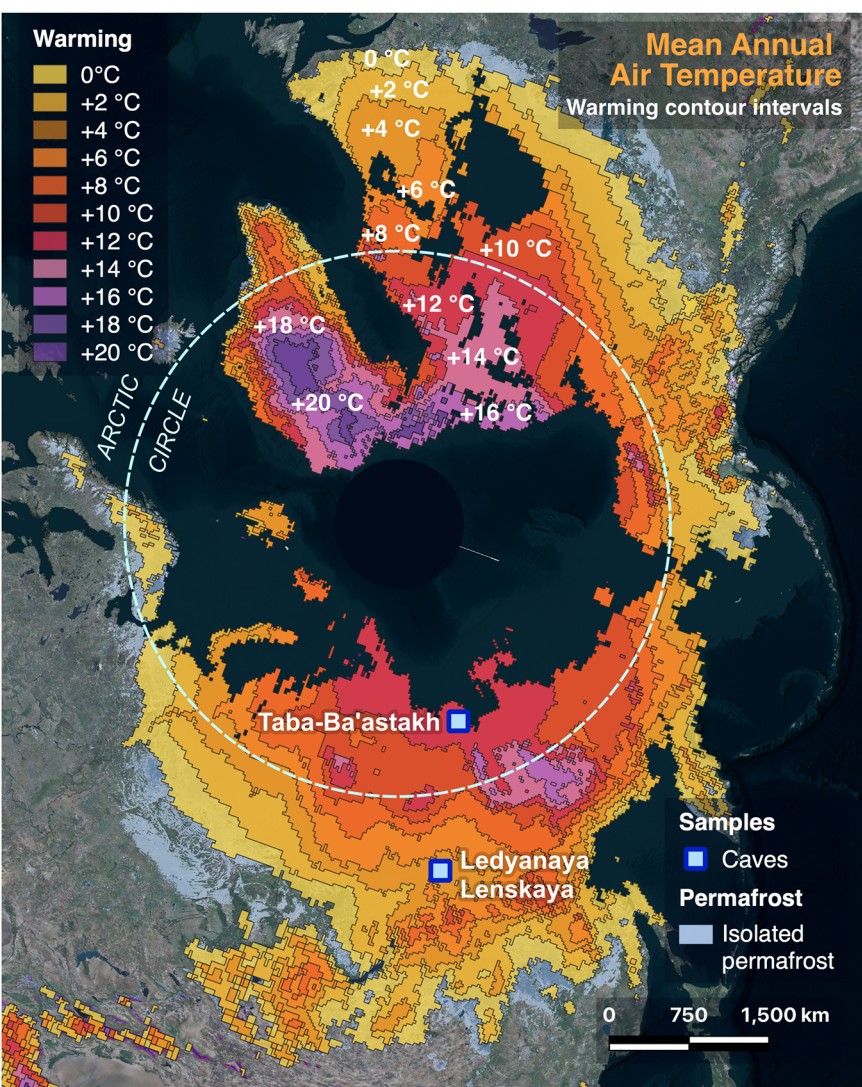

**Fig. 10** | Reduction of the permafrost area[44] in the Northern Hemisphere as a function of an increase in mean annual air temperature (MAAT)[11] at a 2 °C interval. 20 °C+ warming is likely to result in the complete disappearance of permafrost.

Owing to the lack of an independent constraint of the initial $^{234}U/^{238}U$, definitive $^{238}U$-$^{206}Pb$ ages cannot be determined and the scatter in the $^{238}U/^{206}Pb$-$^{208}Pb/^{206}Pb$ data are considered most likely to arise from age-independent variations in the $^{238}U/^{206}Pb$ ratio reflecting a high and variable initial $^{234}U/^{238}U$ ratio, rather than significant real age variation. We therefore take the $^{235}U/^{207}Pb$-$^{208}Pb/^{207}Pb$ isochron age of 8.684 ± 0.086 Ma to be the best age estimate of the entire set of Taba-Ba'astakh speleothems, defining the minimal duration of speleothem deposition period and falling into the late Tortonian. The 8.702 ± 0.378 Ma (2 S.D. $n$ = 19) average model age from the three most radiogenic speleothems also fits into the late Tortonian (Fig. 2).

**Modelling the magnitude of $CO_2$ release from thawing permafrost**

Geospatial data were processed using Quantum Geographic Information System, version 3.22 software. Long-term monthly averaged 2-m surface temperature gridded data from Global Historical Climatology Network−Climate Anomaly Monitoring System (GHCN_CAMS) land stations were obtained for the 1991–2020 period, at a 0.5 × 0.5 degree resolution[11]. This raster file was first averaged for all months to create an annual mean, followed by extracting temperature contours over the Northern Hemisphere at 2-degree Celsius intervals (Fig. 10, Dataset-2).

A soil organic carbon (SOC) model dataset[31] was obtained for the top 0–300 cm of permafrost regions in the Northern Hemisphere (excluding the Tibetan Plateau). For reference, the Tibetan Plateau is estimated to hold approximately 36 (−2.4 to +2.3) PgC[40], a relatively minor contribution to the total northern region. The SOC content for the 0–300 cm depth increment is based on the European Space Agency's Climate Change Initiative Global Land Cover dataset, which includes 651 soil pedons from 6500 samples across 16 study areas in northern permafrost regions with a 300 m/pixel resolution[31]. The total amount of SOC in this 0–300 cm portion of the soil was calculated as 855.42 PgC. Based on the chosen temperature intervals, the SOC dataset was masked to include only the area above a given threshold annual surface temperature. Raster layer statistics performed on these filtered datasets provided the sum of remaining SOC (in kg/m$^{-2}$) after a given amount of warming.

## Data availability

Source data (Datasets 1 and 2) are provided with this paper as supplementary information and can be accessed via zenodo.org, https://doi.org/10.5281/zenodo.15194760. Source data are provided with this paper.

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

## Acknowledgements

We thank the Lena Delta Wildlife Reserve (Ust-Lena Reserve) team, including the Head of the Reserve J. Esaulov and the Head Ranger P. Semionov for hosting the expedition and helping with logistics, permits and other paperwork. Special thanks go to captain A. Karelin and to the sailor G. Achikhasov for operating and navigating the research ship "Orlan". We thank D. Sokolnikov (Speleoclub Arabika, Irkutsk) for his help with expeditions' organization, surveying the caves and collecting samples. We thank V. Stolyarov, G. Stepanov, and J. Samborskiy for filming and documenting the expedition, as well as for assisting with the logistics. We thank A. Svetlakov (Institute of Earth's Crust, Siberian Branch of the Russian Academy of Sciences) for assistance with preparation of the samples. Thanks go also to A. Borshevsky from the Geological Survey of Israel for help with preparation of maps. Any use of trade, firm, or product names is for descriptive purposes only and does not imply endorsement by the U.S. Government. This study was funded by the NERC grant NE-K005057-1 and the Leverhulme Trust project RPG-2020-334.

## Author contributions

V.A. wrote the manuscript, raised the funding, coordinated the research, led the expedition scientifically, searched for the caves and collected the speleothems. M.A.J. performed the U-Pb chronology, including the modification of the method to address the specific characteristics of the speleothems, and participated in manuscript preparation. B.S.F.M. raised the funding, participated in the expedition, participated in research coordination and in manuscript preparation. G.A. led the GIS reconstruction of the permafrost extent in Tortonian, calculated the $CO_2$ release associated with Tortonian-scale warming and participated in manuscript preparation. O.A.V. secured part of the permits, led the expedition in terms of field speleology and logistics, and participated in manuscript preparation. A.I. secured part of the permits, guided the expedition in Lena Delta Nature Reserve, assisted with the expedition logistics, local scientific materials, and participated in manuscript preparation. K.A.M. secured part of the permits, led the samples preparation and initial analyses before shipping them abroad and participated in manuscript preparation. U.S. performed XRD and clumped isotopes paleotemperature analyses and participated in manuscript preparation. L.F.A. assisted to M.A.J. in sample preparation to chronology analyses and participated in manuscript preparation. R.M. prepared the permafrost maps using GIS and participated in manuscript preparation. Sadly, our colleague R.M. did not have a chance to see the paper printed, because he passed away during the final stages of publication. H.G.M. led the Oxford paleoclimate Group during the research, assisting in fundraising, guiding part of the team and participating in manuscript preparation.

## Competing interests

The authors declare no competing interests.
