## [Transparent Peer Review file · Nature Communications]

Arctic speleothems reveal nearly permafrost-free Northern Hemisphere in the late Miocene

Corresponding Author: Dr Sebastian Breitenbach

Version 0:

Reviewer comments:

Reviewer #1

(Remarks to the Author)

This was a clean, clear, and straightforward study on a scientifically interesting and societally relevant topic: Miocene speleothem growth in caves that are today overlain by permafrost. I have very few comments and no concerns, and feel that it is ready for publication.

21 – why is melting (or modeling of melting) of permafrost limited to upper 3 m in a warming scenario? In other words, why was 3 m selected as a benchmark?

24 - anthropogenic greenhouse gases may push atmospheric CO₂ concentrations to levels at which global

28 – with the discussion so far centered on CO₂, it's important to note that a lot of C stored in permafrost is in the form of methane, a more powerful GHG

35 – relict?

61 - above 0°C, which led to permafrost-free conditions

63 - but what about oceanic circulation? Or atmospheric circulation? How do we know that we can evaluate temperatures based on (essentially) GHG forcing, rather than a different heat transport mechanism via the ocean/atmosphere? – aha, you've addressed this on line 124.

70 – although it's cited, briefly mention where these pollen data were obtained – marine records, lakes? Is there enough speleothem material to try looking for pollen in them?

83 – what is this site? A sinkhole? Lake?

91 – I was wondering about the impacts of snow deposition. I'm glad you discussed this.

Reviewer #2

(Remarks to the Author)

Thank you for the opportunity to review the Vaks et al. manuscript titled 'Arctic speleothems reveal Late Miocene permafrost-free Northern Hemisphere'.

The manuscript is reviewed by an 'established reviewer' and an 'ECR co-reviewer'.

Overview

The manuscript presents new speleothem-based research on past Siberian permafrost variability by Vaks et al.. This group has previously been successful in producing high-quality high-profile papers broadly on this theme, notably Vaks et al. (2013), which focussed on the mid-late Quaternary and then Vaks et al. (2020), which extended back to the early Quaternary.

In this particular manuscript, the focus is the Tortonian stage of the Late Miocene, specifically 8.68 ± 0.09 Ma. Another manuscript by the same group, which also presents Siberian speleothems of 8.68 ± 0.09 Ma, is presently in review at *Climate of the Past* (Umbo et al., doi: 10.5194/egusphere-2024-1691) with the title *Speleothem evidence for late Miocene extreme Arctic amplification – an analogue for near future anthropogenic climate change?* At the request of the handling editor, we also consider this ‘sibling’ paper in the wider context of this review.

Overall, documenting Miocene permafrost limits is a major scientific advancement, the paper certainly offers important new results on late-Miocene Siberian climate and the scientific method (mainly dating of speleothems) is sound and reliable. The manuscript does, however, read as if it has been rushed in its preparation a little. There are a number of loose ends that are detailed below, particularly with respect to a more nuanced discussion, but this should be possible to revise for publication.

‘Established reviewer’

Specific comments

Line 10: the authors state that they are using cave carbonates to determine when northern Siberia was last permafrost-free. I find this to be quite a sweeping statement that could be more nuanced. For instance, there are many reasons why the region may be permafrost-free but speleothems did not form – hence the presence of speleothems may not necessarily date to the last time it was permafrost-free. The number of samples analysed is also relatively small, hence there may be an element of sampling bias involved. Finally, northern Siberia is very large, surely the results from this study are only regional in extent and do not cover the whole of northern Siberia?

Line 13 – I suggest this is just ‘late’, and delete ‘middle’

Line 16 – ‘while MAAT at the Siberian Arctic coast were 9°C to $>20^{\circ}\text{C}$ above present’ – if I check the reference that is given (Popova et al., 2012) then it gives MAAT of 9°C but the $>20^{\circ}\text{C}$ relates to the warm months only (not MAAT). The highest MAAT from this region seems to be $+16^{\circ}\text{C}$ from the Omoloy river.

Lines 17-19 – ‘Our findings provide direct observational evidence that warming to temperatures similar to those of the Tortonian period would lead to extensive permafrost thaw along the Siberian Arctic coast’.

I agree there is no doubt that warming to such temperatures would lead to extensive permafrost thaw, however, the statement implies that the temperatures that are being considered here are the 9°C and $>20^{\circ}\text{C}$ from the previous sentence, which comes from another study. The issue is that: (1) the Vaks manuscript doesn’t seem to provide direct observational evidence related to quantitative temperature reconstructions (because it comes from another study), and (2) I am not sure why the authors are not citing their own work on quantitative temperature reconstructions that is in review elsewhere (Umbo et al.).

Line 36 – if I understand correctly, the caves are from one location, hence ‘northern Siberian regions’ may be a little exaggerated

Line 37-38 - “Speleothems found in today’s permafrost regions attest to warmer periods when continuous permafrost was absent to allow water seepage into caves” – it is certainly possible for water to enter caves even when they are in permafrost if there is enough latent heat in the flow and a route for it to take such as an aven or fissure. Water is also required for cryogenic cave carbonates to form, which are a form of speleothem. However, common speleothems such as stalactites, stalagmites and flowstones indeed require a temperature above 0°C so that the water does not freeze. Perhaps this sentence can be revised e.g., “Common speleothems found in today’s permafrost regions attest to warmer periods when continuous permafrost was absent and cave temperatures above freezing”

Line 61-63 – ‘The geographical position of eastern Siberia did not change significantly over the last 80 Ma, suggesting that plate-movement-related latitude changes did not affect climate in the region.’ – A comment on uplift would be appreciated so that we know whether the caves are expected to be in the same position as the time of speleothem deposition.

Lines 80-85 – this comes back to my earlier comment. Whilst this discussion is relevant and very interesting, why do the authors not refer to their own work, which firmly places the temperature estimate between $+6.6$ and $+11.1^{\circ}\text{C}$.

Line 91-93 - “Furthermore, more frequent autumn precipitation as a direct result of incomplete or late sea ice cover during the late Miocene may have contributed to thicker snow cover during winter, insulating the ground and hampering the formation of permafrost” – is it not more likely that this fell as rainfall?

Line 93-95 – “Based on the range of reconstructed temperatures, we estimate the Tortonian northern Siberia featured a MAAT in the range of -3 to $+16^{\circ}\text{C}$, equivalent to an increase of 9 to 28°C from present-day temperatures.”

-This statement is a bit strong, given that the temperature estimates are from other studies. It could be more nuanced to say something like “Based on the range of reconstructed temperatures (refs), Tortonian northern Siberia MAAT is estimated in the range of -3 to $+16^{\circ}\text{C}$, equivalent to an increase of 9 to 28°C from present-day temperatures.”

Line 101 – ‘any speleothem formation implies a MAAT of $>-5^{\circ}\text{C}$, ‘

Please expand and explain further. If the cave air temperature is representative of MAAT, then at 0 to -5°C, cryogenic cave carbonates would form. However, as the authors state above, increased snowfall may insulate the ground – in this case, the cave air temperature will be decoupled from MAAT. This also excludes issues of cave geometry, which I appreciate the authors cannot possibly know with the relics that they have today.

Line 107 - '9-20°C of warming'

-Perhaps specify that this is regional (not global) warming just for clarity

Line 108 onwards – discussion related to permafrost thaw and release of carbon. This is not really my expertise so I cannot comment on the details. It is not clear to me though, why Umbo et al. are already updating this Vaks et al paper

Umbo et al 'Using our new temperature reconstructions, we estimate total potential permafrost derived carbon emissions given future warming similar to that reconstructed for the Tortonian and update the previous calculation from Vaks et al. (in review)' Lines 491-492. Furthermore, figure 3 in this paper is almost identical to figure 7 in Umbo et al. though admittedly with different temperature estimates.

References – please check as they are not all formatted the same, differences in style of dois for example.

References 7 and 31 have doi written twice

Co-reviewer

Specific comments

Abstract (Lines 4-8): The opening lines would benefit from more precise language than "climate change is happening faster in Siberia than globally". What kind of climate change? How much faster? How much permafrost thaw qualifies as 'substantial'?

More importantly, if the goal is to assess the potential for Siberian permafrost thaw with Anthropogenic global warming (AGW) over the next century, the Late Miocene is unlikely to be very helpful, for reasons mostly given later in the manuscript (e.g. lack of Greenland ice sheet, which significantly cools the modern Arctic; partially open Panamanian seaway, altering the NAC/Gulf Stream dynamics; closed Bering Strait; expanded Arctic coastlines and much lower sea-ice limits, enhancing precipitation). For instance, even though CO₂ concentrations were lower than at present, the abrupt post-YD warming during the Early Holocene constitutes a problematic analog to Siberian permafrost degradation from AGW, due to several important differences in the boundary conditions. For example, see discussion and references in Li et al. (2021; <https://www.nature.com/articles/s43247-021-00238-z>).

In short, documenting the Miocene permafrost limits is a major scientific advancement and will greatly elucidate high-latitude climate sensitivities. However, it provides limited information on how much permafrost retreat could happen in response to comparable changes in global MAT today. Under no circumstances does Miocene (vs. Holocene) warming in the Arctic, highly amplified by the paucity of NH ice sheets, describe a reasonable expectation for near-term anthropogenic warming. Hence, I consider the end goal of calculating carbon budgets from permafrost thaw under Miocene climate to be somewhat questionable in 21st-century (or longer) forecasts.

From Popova et al., 2012:

Late Miocene MAAT is near ~12-14°C for coastal northern/eastern Siberia, with CMT around -4 to 0°C, which corresponds to warming of ~26°C (MAAT) and ~30°C (CMT) relative to today (which is already warmer than PI). Using the (slightly lower) Omoloy River data, there is a range of ~20-29°C (MAAT) and ~26°C (CMT) higher than today.

- Why reference these data and then provide such a vague warming estimate of >20°C, especially when it follows such a specific lower range of 9°C? Please consider taking them at face value and provide a specific range, or simply use the upper limit of their reconstruction in all instances (e.g. "Miocene warming along the Arctic coast up to 29°C higher than today").
- Late Miocene precipitation (800-900 mm MAP) is estimated to be slightly higher than the Pliocene (700-800 mm MAP). As far as I can tell, we aren't provided any indication of modern MAP at the cave site, with which to compare these earlier values. But these data are highly important to assessing the relevance of MAAT changes and the expected ecological evolution of a post-permafrost landscape.

Line 29-30: "Mean annual ground warming above 0°C..." Is this supposed to read mean-annual ground temperature (MAGT)? If so, please clarify it in the text. Either way, the discussion would benefit from considering the various thresholds for permafrost cover and the relationship between permafrost, ground temperature, and surface air temperature.

For example, permafrost-stored soil carbon is already susceptible to mobilization at the continuous-discontinuous boundary (around -1.7 ± 0.5°C; Obu et al., 2019), which corresponds (in northern Siberia) to MAAT between -9°C and -6°C (Gruber et al., 2012). The modern study site is already within 5°C of this threshold.

Line 31-32: What is the source of this "considerable uncertainty"? Shouldn't sustained MAGT >0°C do the trick? If so,

calculate the difference between present ground temperatures and 0°C, and that's how much warming is needed. Beyond that, it's only a matter of time (which does have some uncertainty, but as far as I can tell, thaw rates are not addressed in this study).

Line 42-52: The site description is good, but it need not be repeated verbatim in the SI and it lacks two vital components: 1) What is the monthly/seasonal precipitation, locally? 2) What are the ground temperatures near the site? Since the focus is on permafrost extent and degradation, ground temperature is a more important component than air.

U-Pb dating: I congratulate the authors on this stunning geochronological result. Can you clarify in the text, was any attempt made to constrain the range of speleothem-recorded growth? How thin of a snapshot are we talking? I appreciate this may not have been plausible with the sample selection (especially if they're nearly coeval), but if that's the case, there should be mention of how much time feasibly could be recorded by these speleothems.

Line 101: "and the fact that any speleothem formation implies a MAAT of >-5°C..."

Why -5°C? Because of conditions around Botovskaya? Either way, this is an oversimplification of cryospheric processes and thresholds. This sentence could be revised to something like:

"Considering today's MAGT of -XX°C at Taba-Ba'astakh and the fact that any speleothem formation implies a MAGT of at least -2°C (i.e. discontinuous permafrost) but more likely >0°C..."

By relating the discussion to MAAT, the complex relationship between ground and air temperatures is missing, which is important in regions like Siberia with high-amplitude seasonality and substantial winter snowpack (though I appreciate that the insulating effect of snow is briefly mentioned):

Smith, M.W., and D.W. Riseborough 1996. Permafrost Monitoring and Detection of Climate Change, Permafrost and Periglacial Processes, 7, 301–309.

Wright et al., 2003, Regional-scale permafrost mapping using the TTOP ground temperature model

Gruber et al., 2012 - <https://doi.org/10.5194/tc-6-221-2012>

Obu et al., 2019 - <https://doi.org/10.1016/j.earscirev.2019.04.023>

Peng et al., 2024 - <https://doi.org/10.1088/1748-9326/ad30a5>

The Botovskaya Cave region is not necessarily informative of the air-ground insulating effect along the Arctic coastline, because the determining factors are the relative proportion of freezing and thawing degree days (a function of seasonal amplitude), their respective conductivities, and snow-cover days. Hence that air-ground temperature difference of ~5°C near Botovskaya climbs to ~9°C near the Taba-Ba'astakh site. Without specific constraints on changes in winter precipitation and summer-winter amplitude during the Miocene, it is difficult to estimate the magnitude shift in MAAT required to induce permafrost thaw at the modern Arctic coastline (i.e. a +9°C change in MAAT does not necessarily equate to a +9°C change in ground temperature). Please consider incorporating this in the discussion more comprehensively or limit the quantitative estimates to ground-temperature changes as a combined, yet uncertain, function of surface temperature plus winter precipitation. I think the latter almost achieved already but could be improved a little.

Line 114-118: "Estimates by Schuur et al. suggest that 5-15% of this surficial soil organic carbon would be vulnerable to be released as greenhouse gases within decades in the short-term, while longer-term estimates for soil carbon release are poorly constrained. This means that 30-85 PgC will be released into the atmosphere for the conservative estimate of uniform warming of 9°C across Siberia (Fig. 3-C)."

In my opinion, the application of modeling studies by Schuur et al. needs to be refined, and much of the vast literature since 2015 on this topic has been left out.

First, I am not convinced that the cited range of Siberian warming (9-29°C) is realistic for global MAAT forecasts up to +4.5°C, unless you can also remove the Greenland ice sheet in a matter of decades this century.

Second, the estimates for carbon mobilization are based on limited data, hence Schuur et al. caution that several uncertainties exist. These processes have been extensively investigated since Schuur et al. 2015 with updated datasets (refs below), including the find that relatively little of the carbon release during permafrost and peatland warming is from ancient carbon stores, at least during the Early Holocene. Consequently, the manuscript would benefit from refining its discussion to include some of the vast literature post-2015 that has not been cited. Otherwise, it risks oversimplifying cryospheric processes in terms of carbon cycling.

Finally, carbon cycling amid abrupt permafrost degradation is complicated by the specific ecological response, particularly with regard to peatland formation:

Turetsky et al., 2020, <https://www.nature.com/articles/s41561-019-0526-0> (as a start; almost any recent paper by Merritt Turetsky will be highly relevant)

Smith et al., 2004, Siberian peatlands a net carbon sink and global methane source since the Early Holocene

MacDonald et al., 2006, Rapid early development of circumarctic peatlands and atmospheric CH₄ and CO₂ variations.

Alexandrov et al., 2016, The influence of climate on peatland extent in Western Siberia since the Last Glacial Maximum

In short, moisture availability and rates of future warming are important factors in constraining carbon mobilization to atmospheric reservoirs versus terrestrial uptake through peatland expansion. Providing a simple fraction of carbon stored within given isotherms is again misleading, because it belies the complex but well studied dynamics of carbon cycling and storage in permafrost and peatlands. Under what conditions would carbon mobilization to the atmosphere be amplified or mitigated during permafrost thaw in the high Arctic of Siberia?

I hope the authors can develop a more nuanced discussion of these dynamics, because this speleothem-based record will provide an important contribution to the field and to a time and region that is scarcely represented by terrestrial proxy data.

Reviewer #3

(Remarks to the Author)

dear authors,

As requested, I read and reviewed the manuscript titled "Arctic speleothems reveal Late Miocene permafrost-free Northern Hemisphere" by A. Vaks and co-workers submitted for publication in Nature Communications. In their study, the authors present precise ages of speleothems found in a cave system in northern Siberia and argue that the youngest age of these speleothems represents the last time this region was free of permafrost. They use this result together with observations in other caves in the region to determine the degree of warming that is needed to melt the permafrost in the Arctic region. They then use this threshold to inform models of soil organic carbon stored in Siberia under various global warming scenarios to estimate how much carbon could be released from melting permafrost in the future.

This is a well-written manuscript with a clear and important message with relevance for the global climate crisis: The melting of permafrost in future scenarios is one of the most important and most uncertain response mechanisms that could have an accelerating effect on global warming. Therefore, the effort this study makes to better understand the sensitivity to Arctic permafrost systems to warming is of high societal and scientific importance. I find the arguments put forward by the authors to support that the soil above the northern Siberian site under study was at least discontinuously frozen convincing. The way the authors combine observational evidence with modelling, especially embracing the wide array of possible paleotemperatures for the site during the Tortonian, is thoughtful. While I am no expert in interpreting the palynological/vegetation evidence, I find the lines of reasoning put forward by the authors plausible and I am convinced that their conclusions about the sensitivity of the permafrost-stored carbon pool to warming are sound within their given uncertainty.

Two questions occurred to me while reading this work, which I think the authors might be able to answer by expanding upon these subjects in the text. Beyond these more general comments, I detail some additional minor comments which, while not impactful regarding the scientific quality of the study, might help the authors improve their manuscript to more clearly convey its main points. I trust the authors will be able to address these concerns and make the manuscript acceptable for publication after minor revisions.

Seasonal variability

Firstly, given the large temperature seasonality in the region today (see line 45). I wonder whether the presence of speleothem formation at the site in the Tortonian proves that the soil in region was ice-free at this time. Is it not possible that speleothem formation took place during part of the year (e.g. only in summer) and that the soil was completely frozen in the winter? It is not clear from the text or the supplement for how much of the annual temperature cycle the air temperature needs to be above freezing for speleothem formation to occur. It is also not clear whether seasonal temperature variability is taken into account in the soil carbon modelling exercise. Therefore, it remains unclear (to me) how partial, seasonal, or discontinuous freezing of the soil affects the amount of carbon that will be released when the threshold annual air temperature is reached at the soil location. If the observed evidence of speleothem growth can be achieved with a seasonal thawing of the permafrost, I think the title of the manuscript (mentioning a "permafrost-free Northern Hemisphere") is too strong. In that case I would suggest the authors refer in their title to the large amount of carbon that can be released from the Arctic permafrost rather than making statements about the complete disappearance of permafrost in the region, which I believe is not fully supported by the data. The former is, in my opinion, a more important outcome of the study than the (perhaps partly semantic) discussion of whether or not the whole Arctic region was permafrost free.

The Tortonian as an analogue for future climate

The Tortonian features a mean annual global temperature of 4.5 degrees above the pre-industrial temperatures. This is a relatively high temperature scenario when compared to the IPCC scenarios which are currently most in agreement with our global policies and actions. In the introduction, I get the feeling that the authors argue that 2-5 degrees of warming is realistic in the near future (they do not clearly specify exactly when these temperatures could be reached). In my opinion, the manuscript would benefit from a clearer statement on our current global warming scenario and how the Tortonian climate compares to it. I am not arguing that, because the Tortonian was warmer than the climate we are likely to experience in the year 2100, the Tortonian climate is not an interesting case study to better understand the response of permafrost to warming. However, it would be useful for the less-initiated reader to see the Tortonian climate as representative for (if not completely analogous to) a high-warming climate scenario to place the results of this study in that context.

Minor comments

- Line 101: The authors state that speleothem formation requires an MAAT above -5 degrees, but it is not clear what this minimum temperature is based on. Please add some references for this and explain briefly how this threshold is determined.
- Figure 1: The black numbers next to the black stars indicating other speleothem sites are hard to read due to the color contrast. Perhaps the authors can slightly change the background of these numbers to make them stand out a bit more. Otherwise this figure is very clear and conveys a lot of important information in a smart way.
- Figure 2: The diamond indicating the isochron age looks more red than purple to me and is very similar in color to the red

curve indicating the western Arctic temperature evolution. Perhaps the authors can make it stand out a bit more by adding more blue to the purple. In addition, in the text the difference between the two dates and their uncertainties (isochron vs 19 most radiogenic samples) is not clearly explained. In the Methods section (line 110-112), the authors argue that the isochron age to be the most precise age determination. I would suggest the authors either use this ages as representative for the most recent speleothem formation or explain in the main text why two different age estimates are presented to prevent confusion to the more general reader.

References

Climate Action Tracker. 2100 Warming Projections: Emissions and expected warming based on pledges and current policies. <https://climateactiontracker.org/global/temperatures/> (2023).

Reviewer #4

(Remarks to the Author)

Version 1:

Reviewer comments:

Reviewer #1

(Remarks to the Author)

I reviewed an earlier version of this manuscript and offered only minor comments. These have been adequately addressed. In my opinion, the manuscript is acceptable for publication.

Reviewer #2

(Remarks to the Author)

Thank you for the opportunity to read the revised version of Vaks et al., 'Arctic speleothems reveal Late Miocene permafrost-free Northern Hemisphere'. We consider this to be a valuable study of great importance to furthering our understanding of High Latitude climate and permafrost extent. We look forward to seeing it published.

'Established reviewer'

Specific comments

Line 10: the authors state that they are using cave carbonates to determine when northern Siberia was last permafrost-free. I find this to be quite a sweeping statement that could be more nuanced. For instance, there are many reasons why the region may be permafrost-free but speleothems did not form – hence the presence of speleothems may not necessarily date to the last time it was permafrost-free. The number of samples analysed is also relatively small, hence there may be an element of sampling bias involved. Finally, northern Siberia is very large, surely the results from this study are only regional in extent and do not cover the whole of northern Siberia?

- In the response, the reviewers outline a number of valid reasons to address this issue. The abstract has been updated and is more nuanced though I don't see the specific arguments put forward in the rebuttal as having made it into the manuscript (which would be a great help to readers who are not experts in speleothem growth in High Arctic environments). Despite this, I agree with the scientific reasoning and don't have issues with it published as it is.

Line 13 – I suggest this is just 'late', and delete 'middle'

- The authors have addressed this.

Line 16 – 'while MAAT at the Siberian Arctic coast were 9oC to >20oC above present' – if I check the reference that is given (Popova et al., 2012) then it gives MAAT of 9oC but the >20oC relates to the warm months only (not MAAT). The highest MAAT from this region seems to be +16 oC from the Omoloy river.

- The authors have addressed this.

Lines 17-19 – 'Our findings provide direct observational evidence that warming to temperatures similar to those of the Tortonian period would lead to extensive permafrost thaw along the Siberian Arctic coast'.

I agree there is no doubt that warming to such temperatures would lead to extensive permafrost thaw, however, the statement implies that the temperatures that are being considered here are the 9oC and >20oC from the previous sentence, which comes from another study. The issue is that: (1) the Vaks manuscript doesn't seem to provide direct observational evidence related to quantitative temperature reconstructions (because it comes from another study), and (2) I am not sure why the authors are not citing their own work on quantitative temperature reconstructions that is in review elsewhere (Umbo et al.).

- The authors have addressed this.

Line 36 – if I understand correctly, the caves are from one location, hence ‘northern Siberian regions’ may be a little exaggerated

- I am satisfied with this response and thank the authors for explaining the thought process.

Line 37-38 - “Speleothems found in today’s permafrost regions attest to warmer periods when continuous permafrost was absent to allow water seepage into caves” – it is certainly possible for water to enter caves even when they are in permafrost if there is enough latent heat in the flow and a route for it to take such as an aven or fissure. Water is also required for cryogenic cave carbonates to form, which are a form of speleothem. However, common speleothems such as stalactites, stalagmites and flowstones indeed require a temperature above 0°C so that the water does not freeze. Perhaps this sentence can be revised e.g., “Common speleothems found in today’s permafrost regions attest to warmer periods when continuous permafrost was absent and cave temperatures above freezing”

- The authors have addressed this.

Line 61-63 – ‘The geographical position of eastern Siberia did not change significantly over the last 80 Ma, suggesting that plate-movement-related latitude changes did not affect climate in the region.’ – A comment on uplift would be appreciated so that we know whether the caves are expected to be in the same position as the time of speleothem deposition.

- It is a little buried and could be more upfront, but the authors have addressed this.

Lines 80-85 – this comes back to my earlier comment. Whilst this discussion is relevant and very interesting, why do the authors not refer to their own work, which firmly places the temperature estimate between +6.6 and +11.1 °C.

- The authors have addressed this.

Line 91-93 - “Furthermore, more frequent autumn precipitation as a direct result of incomplete or late sea ice cover during the late Miocene may have contributed to thicker snow cover during winter, insulating the ground and hampering the formation of permafrost” – is it not more likely that this fell as rainfall?

- The authors have addressed this.

Line 93-95 – “Based on the range of reconstructed temperatures, we estimate the Tortonian northern Siberia featured a MAAT in the range of -3 to +16 °C, equivalent to an increase of 9 to 28°C from present-day temperatures.”

-This statement is a bit strong, given that the temperature estimates are from other studies. It could be more nuanced to say something like “Based on the range of reconstructed temperatures (refs), Tortonian northern Siberia MAAT is estimated in the range of -3 to +16 °C, equivalent to an increase of 9 to 28°C from present-day temperatures.”

-The authors have addressed this.

Line 101 – ‘any speleothem formation implies a MAAT of >-5°C,’

Please expand and explain further. If the cave air temperature is representative of MAAT, then at 0 to -5°C, cryogenic cave carbonates would form. However, as the authors state above, increased snowfall may insulate the ground – in this case, the cave air temperature will be decoupled from MAAT. This also excludes issues of cave geometry, which I appreciate the authors cannot possibly know with the relics that they have today.

- The authors have addressed this.

Line 107 - ‘9-20°C of warming’. Perhaps specify that this is regional (not global) warming just for clarity

- The authors have addressed this.

Line 108 onwards – discussion related to permafrost thaw and release of carbon. This is not really my expertise so I cannot comment on the details. It is not clear to me though, why Umbo et al. are already updating this Vaks et al paper

Umbo et al ‘Using our new temperature reconstructions, we estimate total potential permafrost derived carbon emissions given future warming similar to that reconstructed for the Tortonian and update the previous calculation from Vaks et al. (in review)’ Lines 491-492. Furthermore, figure 3 in this paper is almost identical to figure 7 in Umbo et al. though admittedly with different temperature estimates.

- The authors have addressed this.

References – please check as they are not all formatted the same, differences in style of doi for example. References 7 and 31 have doi written twice

- The authors have addressed this.

Co-reviewer

Specific comments

Abstract (Lines 4-8): The opening lines would benefit from more precise language than "climate change is happening faster in Siberia than globally". What kind of climate change? How much faster? How much permafrost thaw qualifies as 'substantial'?

- The authors have addressed this comment.

More importantly, if the goal is to assess the potential for Siberian permafrost thaw with Anthropogenic global warming (AGW) over the next century, the Late Miocene is unlikely to be very helpful, for reasons mostly given later in the manuscript (e.g. lack of Greenland ice sheet, which significantly cools the modern Arctic; partially open Panamanian seaway, altering the NAC/Gulf Stream dynamics; closed Bering Strait; expanded Arctic coastlines and much lower sea-ice limits, enhancing precipitation). For instance, even though CO₂ concentrations were lower than at present, the abrupt post-YD warming during the Early Holocene constitutes a problematic analog to Siberian permafrost degradation from AGW, due to several important differences in the boundary conditions. For example, see discussion and references in Li et al. (2021; <https://www.nature.com/articles/s43247-021-00238-z>).

In short, documenting the Miocene permafrost limits is a major scientific advancement and will greatly elucidate high-latitude climate sensitivities. However, it provides limited information on how much permafrost retreat could happen in response to comparable changes in global MAT today. Under no circumstances does Miocene (vs. Holocene) warming in the Arctic, highly amplified by the paucity of NH ice sheets, describe a reasonable expectation for near-term anthropogenic warming. Hence, I consider the end goal of calculating carbon budgets from permafrost thaw under Miocene climate to be somewhat questionable in 21st-century (or longer) forecasts.

- The authors have sufficiently addressed my concerns in the updated text. I appreciate the clarifications on the difference in boundary conditions.

From Popova et al., 2012:

Late Miocene MAAT is near ~12-14°C for coastal northern/eastern Siberia, with CMT around -4 to 0°C, which corresponds to warming of ~26°C (MAAT) and ~30°C (CMT) relative to today (which is already warmer than PI). Using the (slightly lower) Omoloy River data, there is a range of ~20-29°C (MAAT) and ~26°C (CMT) higher than today.

• Why reference these data and then provide such a vague warming estimate of >20°C, especially when it follows such a specific lower range of 9°C? Please consider taking them at face value and provide a specific range, or simply use the upper limit of their reconstruction in all instances (e.g. "Miocene warming along the Arctic coast up to 29°C higher than today").

- The authors have addressed this concern with citation of paleo-T estimates from Umbo et al.

• Late Miocene precipitation (800-900 mm MAP) is estimated to be slightly higher than the Pliocene (700-800 mm MAP). As far as I can tell, we aren't provided any indication of modern MAP at the cave site, with which to compare these earlier values. But these data are highly important to assessing the relevance of MAAT changes and the expected ecological evolution of a post-permafrost landscape.

- The authors have addressed this comment with added MAP data.

Line 29-30: "Mean annual ground warming above 0°C..." Is this supposed to read mean-annual ground temperature (MAGT)? If so, please clarify it in the text. Either way, the discussion would benefit from considering the various thresholds for permafrost cover and the relationship between permafrost, ground temperature, and surface air temperature.

- The authors have addressed this comment and now clearly distinguish the role of ground temperature throughout the manuscript.

For example, permafrost-stored soil carbon is already susceptible to mobilization at the continuous-discontinuous boundary (around $-1.7 \pm 0.5^\circ\text{C}$; Obu et al., 2019), which corresponds (in northern Siberia) to MAAT between -9°C and -6°C (Gruber et al., 2012). The modern study site is already within 5°C of this threshold.

Line 31-32: What is the source of this "considerable uncertainty"? Shouldn't sustained MAGT >0°C do the trick? If so, calculate the difference between present ground temperatures and 0°C, and that's how much warming is needed. Beyond that, it's only a matter of time (which does have some uncertainty, but as far as I can tell, thaw rates are not addressed in this study).

Line 42-52: The site description is good, but it need not be repeated verbatim in the SI and it lacks two vital components: 1) What is the monthly/seasonal precipitation, locally? 2) What are the ground temperatures near the site? Since the focus is on permafrost extent and degradation, ground temperature is a more important component than air.

- The authors have addressed this fully.

U-Pb dating: I congratulate the authors on this stunning geochronological result. Can you clarify in the text, was any attempt made to constrain the range of speleothem-recorded growth? How thin of a snapshot are we talking? I appreciate this may not have been plausible with the sample selection (especially if they're nearly coeval), but if that's the case, there should be mention of how much time feasibly could be recorded by these speleothems.

- The authors have answered this well. The updated text provides a much clearer picture of the chronology and potential time span represented in speleothem collections.

Line 101: "and the fact that any speleothem formation implies a MAAT of $>-5^{\circ}\text{C}$..."

Why -5°C ? Because of conditions around Botovskaya? Either way, this is an oversimplification of cryospheric processes and thresholds. This sentence could be revised to something like:

"Considering today's MAGT of $-XX^{\circ}\text{C}$ at Taba-Ba'astakh and the fact that any speleothem formation implies a MAGT of at least -2°C (i.e. discontinuous permafrost) but more likely $>0^{\circ}\text{C}$..."

- I appreciate the revised discussion in lines 68–73, which shows that paleo-T estimates from 6.6 – 11.1°C preclude the possibility even of discontinuous permafrost, even accounting for uncertainties. However, I feel the authors revert to an oversimplification of these clumped-isotope data, particularly in lines 81–84, where they conflate them with MAAT:

- "Clumped isotope paleotemperatures from Taba-Ba'astakh speleothems show MAAT of $+6.6^{\circ}\text{C}$ - $+11.1^{\circ}\text{C}$, indicating a warming of 18.9°C to 23.4°C above present, considering the difference between the Tortonian MAAT and present day MAAT of -12.3°C ."

- The clumped isotope approach specifically provides an estimate of the fluid temperature during speleothem growth. This temperature is certainly related to, but is not the same as, MAAT at the surface. Drip-water temperature reflects the mean ground and cave-air temperatures at depth, which are insulated by the extent and length of snow cover (among other factors, discussed in my original comments below). High-latitude cave sites, particularly in Siberia, almost invariably exhibit warmer internal temperatures than the mean surface air temperature, some by at least 4°C . It is entirely plausible that a Late Miocene cave temperature of 6.6°C corresponded to a MAAT much closer to 0°C (but certainly less than 6.6°C).

- Calculating a precise Nival offset for the site would be inconsequential to your conclusions herein. It's also impossible to constrain from your dataset. What matters most is that the cave/ground temperature is likely to be at least $\sim 6.6^{\circ}\text{C}$, which is well above 0°C and thus permafrost-free. Given the importance of this study, I strongly feel it is necessary to avoid such an oversimplification as equating the clumped isotope estimates directly with MAAT.

- I recommend and would appreciate a simple statement in the methods explaining that the authors assume cave temperature and MAAT to be the same and why they think it's reasonable to assume this.

By relating the discussion to MAAT, the complex relationship between ground and air temperatures is missing, which is important in regions like Siberia with high-amplitude seasonality and substantial winter snowpack (though I appreciate that the insulating effect of snow is briefly mentioned):

Smith, M.W., and D.W. Riseborough 1996. Permafrost Monitoring and Detection of Climate Change, Permafrost and Periglacial Processes, 7, 301–309.

Wright et al., 2003, Regional-scale permafrost mapping using the TTOP ground temperature model

Gruber et al., 2012 - <https://doi.org/10.5194/tc-6-221-2012>

Obu et al., 2019 - <https://doi.org/10.1016/j.earscirev.2019.04.023>

Peng et al., 2024 - <https://doi.org/10.1088/1748-9326/ad30a5>

The Botovskaya Cave region is not necessarily informative of the air-ground insulating effect along the Arctic coastline, because the determining factors are the relative proportion of freezing and thawing degree days (a function of seasonal amplitude), their respective conductivities, and snow-cover days. Hence that air-ground temperature difference of $\sim 5^{\circ}\text{C}$ near Botovskaya climbs to $\sim 9^{\circ}\text{C}$ near the Taba-Ba'astakh site. Without specific constraints on changes in winter precipitation and summer-winter amplitude during the Miocene, it is difficult to estimate the magnitude shift in MAAT required to induce permafrost thaw at the modern Arctic coastline (i.e. a $+9^{\circ}\text{C}$ change in MAAT does not necessarily equate to a $+9^{\circ}\text{C}$ change in ground temperature). Please consider incorporating this in the discussion more comprehensively or limit the quantitative estimates to ground-temperature changes as a combined, yet uncertain, function of surface temperature plus winter precipitation. I think the latter almost achieved already but could be improved a little.

- The authors have addressed my concerns about the reference to Botovskaya Cave.

Line 114-118: "Estimates by Schuur et al. suggest that 5-15% of this surficial soil organic carbon would be vulnerable to be released as greenhouse gases within decades in the short-term, while longer-term estimates for soil carbon release are poorly constrained. This means that 30-85 PgC will be released into the atmosphere for the conservative estimate of uniform warming of 9°C across Siberia (Fig. 3-C)."

- The authors have addressed this comment. I appreciate the updated references and discussion.

In my opinion, the application of modeling studies by Schuur et al. needs to be refined, and much of the vast literature since 2015 on this topic has been left out. Consequently, the manuscript oversimplifies cryospheric processes also in terms of carbon cycling.

First, I am not convinced that the cited range of Siberian warming (9-29°C) is realistic for global MAAT forecasts up to +4.5°C, unless you can also remove the Greenland ice sheet in a matter of decades this century.

- This comment has been addressed. I think it is now sufficiently communicated that the Miocene analogy is a long-term potential response to high warming scenarios, rather than a near-term expectation.

Second, the estimates for carbon mobilization are based on limited data, hence Schuur et al. caution that several uncertainties exist. These processes have been extensively investigated since Schuur et al. 2015 with updated datasets (refs below), including the find that relatively little of the carbon release during permafrost and peatland warming is from ancient carbon stores, at least during the Early Holocene. Consequently, the manuscript would benefit from refining its discussion to include some of the vast literature post-2015 that has not been cited. Otherwise, it risks oversimplifying cryospheric processes in terms of carbon cycling.

Finally, carbon cycling amid abrupt permafrost degradation is complicated by the specific ecological response, particularly with regard to peatland formation:

Turetsky et al., 2020, <https://www.nature.com/articles/s41561-019-0526-0> (as a start; almost any recent paper by Merritt Turetsky will be highly relevant)

Smith et al., 2004, Siberian peatlands a net carbon sink and global methane source since the Early Holocene

MacDonald et al., 2006, Rapid early development of circumarctic peatlands and atmospheric CH₄ and CO₂ variations.

Alexandrov et al., 2016, The influence of climate on peatland extent in Western Siberia since the Last Glacial Maximum

In short, moisture availability and rates of future warming are important factors in constraining carbon mobilization to atmospheric reservoirs versus terrestrial uptake through peatland expansion. Providing a simple fraction of carbon stored within given isotherms is again misleading, because it belies the complex but well studied dynamics of carbon cycling and storage in permafrost and peatlands. Under what conditions would carbon mobilization to the atmosphere be amplified or mitigated during permafrost thaw in the high Arctic of Siberia?

I hope the authors can develop a more nuanced discussion of these dynamics, because this speleothem-based record will provide an important contribution to the field and to a time and region that is scarcely represented by terrestrial proxy data.

-The authors have addressed this comment in their response and with the updated references.

- Additional comments:

- Line 192-193, Methods: Did you mean to write Mean Ground Annual Temperatures (MGAT) instead of Mean Annual Ground Temperatures (MAGT)? It is now inconsistent with the main text.

Reviewer #3

(Remarks to the Author)

dear authors,

Thank you for asking my second opinion on the manuscript by Vaks et al. As requested, I have gone over the manuscript and the author's rebuttal again and I must conclude that the authors did a great job responding to my points. Their explanation of the impact of changes in temperature seasonality on the permafrost is clear, and makes sense to me given the definition of permafrost used by the authors. While I still think that it would be helpful to clarify exactly for which type of future climate scenario the Tortonian is an analogue, I sympathize with the fact that this might be too much detail for the discussion in this manuscript and agree with the authors' solution in lines 132-134. I also appreciate the improvements to Figures 2 and 3 and the explanation of the age constraints of the speleothem samples. Given the extensive replies of the authors to the other reviewers, I believe this manuscript is now ready for publication in Nature Communications and would like to congratulate the authors on their nice study.

Reviewer #4

(Remarks to the Author)

Dear reviewers and Editor,

We would like to thank the reviewers and the editor for their fruitful comments and suggestions that significantly improved the quality of our manuscript. We revised the manuscript and our answers to all reviewer comments are below:

Reviewer #1 (Remarks to the Author):

The reviewer:

This was a clean, clear, and straightforward study on a scientifically interesting and societally relevant topic: Miocene speleothem growth in caves that are today overlain by permafrost. I have very few comments and no concerns, and feel that it is ready for publication.

We are grateful for the reviewer's positive opinion about the manuscript.

21 – why is melting (or modeling of melting) of permafrost limited to upper 3 m in a warming scenario? In other words, why was 3 m selected as a benchmark?

Our reply:

The surface layer of permafrost is the most carbon-rich and the most vulnerable to interaction and exchange with the atmosphere on shorter (seasonal to centennial) time scales, but you are correct – there is a pool of deeper SOC that we do not include in our estimates. The SOC dataset we use by Palmtag et al., 2022 (<https://doi.org/10.5194/essd-14-4095-2022>, ref 29 in the manuscript), is limited to 3 m depth based on the field sampling method limitations (many soil locations sampled across the Arctic to determine carbon content) and shows that the top meter of soil contains almost half of the total carbon. To show how challenging it is to achieve even that depth, the first estimate for SOC beyond 1 m depth in permafrost regions wasn't compiled until 2013 (Hugelius et al., 2013, <https://doi.org/10.5194/essd-5-393-2013>), and even deeper (> 3 m) SOC stocks are very challenging to quantify (Treat et al., 2024, <https://doi.org/10.1029/2023JG007638>, ref 30 in manuscript). That said, deeper (>3 m) SOC contributes around 25% of the total SOC (e.g., Hugelius et al., 2014, doi:10.5194/bg-11-6573-2014). Therefore, our SOC estimates are likely conservative, as they do not extend below 3 m, and we have noted this in the revised manuscript:

“Therefore, most conservatively 42.77-128.31 PgC will be released into the atmosphere for uniform warming of 18.9°C and higher across the Northern hemisphere permafrost region (Fig. 3-B)” (lines 123-124).

The reviewer:

24 - anthropogenic greenhouse gases may push atmospheric CO₂ concentrations to levels at which global

Our reply:

We are unsure what should be noted or changed here – this is a direct quote from line 24 in the old manuscript, but we think it may tie to the next comment.

The reviewer:

28 – with the discussion so far centered on CO₂, it's important to note that a lot of C stored in permafrost is in the form of methane, a more powerful GHG

Our reply:

Yes, these two greenhouse gas forms are now both mentioned in lines 19 and 117. Initially we didn't highlight CH₄, and the SOC dataset we use simply sums the total soil organic carbon in the top three meters – this carbon could of course be emitted as CO₂ or CH₄ (many modelling studies are attempting to estimate what the proportion of the release is in each form, and a good overview of the current understanding can be found in Treat et al., 2024 (<https://doi.org/10.1029/2023JG007638>, ref 30 in this manuscript).

The reviewer:

35 – relict?

Our reply:

We corrected to “relict” (line 32)

The reviewer:

61 - above 0oC, which led to permafrost-free conditions

Our reply:

We wrote “MAGT above 0°C” (lines 64-65) because even with MAAT between 0°C and -6 °C (or even -8°C in some cases) permafrost can be discontinuous, allowing speleothem deposition.

The reviewer:

63 - but what about oceanic circulation? Or atmospheric circulation? How do we know that we can evaluate temperatures based on (essentially) GHG forcing, rather than a different heat transport mechanism via the ocean/atmosphere? – aha, you've addressed this on line 124.

Our reply:

We addressed it in lines 129-132 indeed.

The reviewer:

70 – although it's cited, briefly mention where these pollen data were obtained – marine records, lakes? Is there enough speleothem material to try looking for pollen in them?

Our reply:

See next comment below.

The reviewer:

83 – what is this site? A sinkhole? Lake?

Our reply:

These are all Miocene continental fluvial/lacustrine sediments as mentioned by the references

inside Pound et al. (2012, Appendices, ref 24 in the manuscript). We added “from continental fluvial/lacustrine sediments” in line 78, “sediments from the Omoloy River” in lines 85 and “sediments from nearby Temmirdekh-Khaja” in lines 87.

The reviewer: 91 – I was wondering about the impacts of snow deposition. I’m glad you discussed this.

Our reply:

In revised version of the paper we cite the Mean Annual Air Temperatures (MAAT) determined from the clumped isotope record of the same Taba-Ba’astakh speleothems (Umbo et al 2024, ref 4). These MAAT are estimated to be +6.6°C - +11.1°C, which makes autumn snow unlikely. These MAAT estimates are similar to the ones shown by Popova et al. (2012, ref 26) and Pound et al. (2012, ref 24) from vegetation/pollen data. The latter two studies (and the references within them) determined Tortonian winter temperatures of -4°C to +1°C, and summer temperatures around +22°C (described in lines 81-89). Therefore the climate was temperate and humid like in Central Europe today, winter ground freezing was minor and couldn’t affect the growth of speleothems.

We removed the snowpack thickness discussion, writing instead:

“These lines of evidence imply a temperate climate and an ice-free Arctic Ocean (at least during summer) in the Tortonian (ref 27) (Fig. 2). Under such climate conditions, only brief, if any, ground freezing in mid-winter would occur, not hampering speleothem growth that was enhanced by high Miocene mean-annual precipitation of 800-900 mm (ref 26).” (lines 94-97).

Reviewer #2 (Remarks to the Author):

The reviewer:

Thank you for the opportunity to review the Vaks et al. manuscript titled ‘Arctic speleothems reveal Late Miocene permafrost-free Northern Hemisphere’.

The manuscript is reviewed by an ‘established reviewer’ and an ‘ECR co-reviewer’.

Overview

The manuscript presents new speleothem-based research on past Siberian permafrost variability by Vaks et al.. This group has previously been successful in producing high-quality high-profile papers broadly on this theme, notably Vaks et al. (2013), which focussed on the mid-late Quaternary and then Vaks et al. (2020), which extended back to the early Quaternary.

In this particular manuscript, the focus is the Tortonian stage of the Late Miocene, specifically 8.68±0.09 Ma. Another manuscript by the same group, which also presents Siberian speleothems of 8.68±0.09 Ma, is presently in review at *Climate of the Past* (Umbo et al., 2024, doi: 10.5194/egusphere-2024-1691) with the title *Speleothem evidence for late Miocene extreme Arctic amplification – an analogue for near future anthropogenic climate change?* At the request of the handling editor, we also consider this ‘sibling’ paper in the wider context of this review.

Overall, documenting Miocene permafrost limits is a major scientific advancement, the paper certainly offers important new results on late-Miocene Siberian climate and the scientific method (mainly dating of speleothems) is sound and reliable. The manuscript does, however, read as if it has been rushed in its preparation a little. There are a number of loose ends that are detailed below, particularly with respect to a more nuanced discussion, but this should be possible to revise for publication.

'Established reviewer'

Specific comments:

The reviewer:

Line 10: the authors state that they are using cave carbonates to determine when northern Siberia was last permafrost-free. I find this to be quite a sweeping statement that could be more nuanced. For instance, there are many reasons why the region may be permafrost-free but speleothems did not form – hence the presence of speleothems may not necessarily date to the last time it was permafrost-free.

Our Reply:

Speleothems don't grow in the carbonate cave when the following conditions occur:

- i) Severe aridity with insufficient effective infiltration to allow water to dissolve the soil CO₂, then dissolve enough carbonate and reach the cave. The proximity to Arctic Ocean moisture source makes this scenario very unlikely. Even today the amount of precipitation in the area is ~300 mm (ref. 18) which is sufficient for speleothem deposition, especially in a cold/temperate climate with little evaporation (Vaks et al. 2010, Quaternary Science Reviews 29, 19-20, 2647-2662, <https://doi.org/10.1016/j.quascirev.2010.06.014>). In the Miocene the amount of precipitation has been estimated to be 700-900 mm (ref. 26), definitely sufficient to allow speleothem deposition because the mean temperatures were +6.6°C - +11.1°C and the ground was not continuously frozen (Umbo et al., 2024, ref 4).
- ii) Clogging of water seepage conduits and a lateral shift of the drip spot at the ceiling so that a stalagmite is no longer fed by the incoming drip. The speleothems were collected from multiple locations along a ca. 500 m long cliff, therefore simultaneous clogging of all water conduits is unlikely;
- iii) Continuous permafrost that leads to a complete still stand of carbonate deposition due to lack of fluid water. This is the only factor that prevents speleothem deposition in Siberia (and other periglacial regions) today. There was no permafrost in Tortonian because of MAAT between 6.6°C and 11.1°C (Umbo et al. 2024, ref 4). Temperatures of Popova et al. 2012 (ref 26) fall in the similar range.

The reviewer:

“The number of samples analyzed is also relatively small, hence there may be an element of sampling bias involved.”

Our reply:

There are 14 speleothems collected at many different sites along a 500 m long cliff (See Figs. 5-8). Each of these speleothems grew in different location and sometimes slightly different altitude, receiving water from different cracks/sources. In this case even if there were impermeable layers somewhere, it is very unlikely that they affected every crack and speleothem. Fourteen speleothems from different locations should be considered to be a relatively large collection; most speleothem studies use far fewer samples (and in many cases only 1 stalagmite).

We acknowledge that zones of permafrost in cooler regions closer to the pole would still remain (see Fig. 3). Therefore, we have added nuance to this statement in the title and in the abstract. The title now reads: “Arctic speleothems reveal **nearly** permafrost-free Northern Hemisphere in the Late Miocene”, and the abstract now reads “Here we use cave carbonates (speleothems) from a northern Siberian cave (Fig. 1) to determine when the Northern Hemisphere was mostly permafrost-free.” (Lines 7-8).

The reviewer:

“Finally, northern Siberia is very large, surely the results from this study are only regional in extent and do not cover the whole of northern Siberia?”

Our reply:

The location of the research site is 72°N latitude, with a modern -12.3°C MAAT isotherm. We can at least suggest that if there is no continuous permafrost on this isotherm, there is also no continuous permafrost anywhere south of this isotherm, which when examining GIS reanalysis data shows that this is the most of Siberia (maybe except for high mountainous regions). Paleotemperature estimates from Umbo et al. (2024) indicate a Miocene MAAT of +6.6°C - +11.1°C (ref 4), and our GIS analysis suggests that almost the entire Tortonian Northern Hemisphere was permafrost-free in these conditions.

The reviewer:

Line 13 – I suggest this is just ‘late’, and delete ‘middle’

Our reply:

In the abstract we now mention only the “Late Tortonian stage” (line 10), while the explanation that Tortonian is a part of late Miocene appears in the manuscript (line 60).

The reviewer:

Line 16 – ‘while MAAT at the Siberian Arctic coast were 9oC to >20oC above present’ – if I check the reference that is given (Popova et al., 2012) then it gives MAAT of 9oC but the >20oC relates to the warm months only (not MAAT). The highest MAAT from this region seems to be +16 oC from the Omoloy river.

Our reply:

We changed this sentence citing Umbo et al. (2024) (ref 4) where the MAAT is estimated to be +6.6°C - +11.1°C as follows: "Paleotemperatures reconstructed from speleothems show that mean annual air temperatures (MAAT) in the region were +6.6°C-+11.1°C (ref 4), at a time when global MAAT were ~4.5°C higher than modern (ref 5)." Lines 11-13.

The reviewer:

Lines 17-19 – 'Our findings provide direct observational evidence that warming to temperatures similar to those of the Tortonian period would lead to extensive permafrost thaw along the Siberian Arctic coast'.

I agree there is no doubt that warming to such temperatures would lead to extensive permafrost thaw, however, the statement implies that the temperatures that are being considered here are the 9oC and >20oC from the previous sentence, which comes from another study. The issue is that: (1) the Vaks manuscript doesn't seem to provide direct observational evidence related to quantitative temperature reconstructions (because it comes from another study), and (2) I am not sure why the authors are not citing their own work on quantitative temperature reconstructions that is in review elsewhere (Umbo et al.).

Our reply:

We agree with the reviewer. Here we can better support our conclusions based on Umbo et al. (2024) (ref 4) paleotemperatures based on clumped isotopes from the same speleothems (MAAT= +6.6 - +11.1°C). Such temperatures support the argument in lines 13-16. We did not cite the Umbo et al. study previously as that paper was not submitted at the time of the submission of this manuscript.

The reviewer:

Line 36 – if I understand correctly, the caves are from one location, hence 'northern Siberian regions' may be a little exaggerated

Our reply:

The GIS reconstruction of how MAAT will change spatially if the present-day MAAT of -12.3 would increase to +6.6 - +11.1°C (Umbo et al. 2024, ref 4) can be extrapolated to the entire Northern Hemisphere, assuming a similar distribution of isothermals across Eurasia in the middle Miocene. Although Taba-Ba'astakh is "one site", the 14 speleothems were taken along a ca. 500 m long cliff at various places (See Figs. 5-8). This means that the water these carbonates grew from was coming from the surface through different cracks/passages taking different routes through the epikarst overburden. If 14 speleothems in different places show roughly the same depositional period, it follows that their deposition was caused by climatic conditions and not by any kind of artifact.

Although such GIS reconstruction of permafrost reduction may look speculative, the clumped-isotope paleotemperature, paleovegetation and SST data presented in the manuscript together show that polar Northern Hemisphere in Tortonian was ~20°C warmer than present. Therefore,

permafrost GIS reconstruction has a high potential to show a real climatic situation for polar Northern Hemisphere in Tortonian.

The reviewer:

Line 37-38 - "Speleothems found in today's permafrost regions attest to warmer periods when continuous permafrost was absent to allow water seepage into caves" – it is certainly possible for water to enter caves even when they are in permafrost if there is enough latent heat in the flow and a route for it to take such as an aven or fissure. Water is also required for cryogenic cave carbonates to form, which are a form of speleothem. However, common speleothems such as stalactites, stalagmites and flowstones indeed require a temperature above 0°C so that the water does not freeze. Perhaps this sentence can be revised e.g., "Common speleothems found in today's permafrost regions attest to warmer periods when continuous permafrost was absent and cave temperatures above freezing"

Our reply:

Water does not enter the caves when permafrost is continuous. If water has enough latent heat to seep into the cave, the surrounding permafrost is discontinuous. Cryogenic carbonates form when water seeps into the caves through the unfrozen ceiling (or along passages), but on the floor the temperatures are sub-zero (e.g., due to seasonal cooling via cave air ventilation), not necessary permafrost (Munroe, J. *et al.* Cryogenic cave carbonate and implications for thawing permafrost at Winter Wonderland Cave, Utah, USA. *Scientific Reports* **11**, 6430, doi:10.1038/s41598-021-85658-9 (2021).). This situation does not occur in continuous permafrost because liquid water cannot seep into the cave under such conditions.

We changed the sentence: "Vadose speleothems found in today's permafrost regions attest to warmer periods when continuous permafrost was absent and cave temperatures were above freezing, allowing water seepage into caves (refs 15, 16)" Lines 33-35.

The reviewer:

Line 61-63 – 'The geographical position of eastern Siberia did not change significantly over the last 80 Ma, suggesting that plate-movement-related latitude changes did not affect climate in the region.' – A comment on uplift would be appreciated so that we know whether the caves are expected to be in the same position as the time of speleothem deposition.

Our reply:

There is evidence of mid-Miocene compressional tectonic activity in the northern part of the Verkhojansk Range where the Taba-Ba'astakh site is located (Imaev et al., 2018 and references within, in Russian, ref 38). It is possible that this compression led to the uplift of the caves above the groundwater level, enabling the growth of vadose speleothems in the late-Miocene. In the Methods section we wrote: "In terms of vertical movements, during the middle Miocene, the region underwent compression that contributed to the uplift of the mountain ridges we see today (ref 38)." Lines 206-208

The reviewer:

Lines 80-85 – this comes back to my earlier comment. Whilst this discussion is relevant and very interesting, why do the authors not refer to their own work, which firmly places the temperature estimate between +6.6 and +11.1°C.

Our reply:

The reviewer is correct, and we changed the manuscript and GIS analysis accordingly:

“The minimum MAAT range that leads to discontinuous permafrost and enables water penetration into the caves is 0°C to -6°C (ref 23), however, clumped isotope analyses performed on four Taba-Ba’astakh speleothems show paleotemperatures of +6.6°C - +11.1°C (ref 4). If MAAT was in this range, it follows that the climate was temperate and permafrost was absent at this site and further south, indicating that most of the Siberian landmass and likely similar regions in the Northern Hemisphere were permafrost-free when speleothems formed at Taba-Ba’astakh.” (Lines 68-73).

Clumped isotope paleotemperatures from Taba-Ba’astakh speleothems show MAAT of +6.6°C - +11.1°C (ref 4), indicating a warming of 18.9°C to 23.4°C above present, considering the difference between the Tortonian MAAT and present day MAAT of -12.3°C. (Lines 81-84)

“Considering today’s MAAT of -12.3°C at Taba-Ba’astakh, we chose an increase of 18.9°C above present in polar regions of the Northern Hemisphere, to MAAT of +6.6°C obtained from clumped isotope measurement of Taba-Ba’astakh speleothems (ref 4). This is our conservative estimate of warming during the Tortonian (Fig. 3-A and 3-B, Dataset-2, Methods).” (Lines 103-106)

“Relating our paleoclimate data from the Tortonian to future warming scenarios in Siberia suggests that most to all of the surficial permafrost would thaw with 18.9°C of warming, with only 1% of present-day permafrost areas remaining as discontinuous permafrost (the purple regions in northern Greenland and Ellesmere Island in Fig. 3-A). Model scenarios of 20°C warming or higher (relative to present) (ref 4) show that no surficial permafrost would remain in the Northern Hemisphere (see Methods). If all near-surface permafrost existing today in the Northern Hemisphere would thaw, an estimated ~855.42 PgC from the top 3 m of soil would become vulnerable for mobilization (Fig. 3-B). This estimate is an underestimate of the total vulnerable carbon, considering that additional organic carbon seated deeper than 3 m might be mobilized, although that the relative contribution of this deeper carbon stock to the total release remains poorly constrained.” (Lines 107-116).

The reviewer:

Line 91-93 - “Furthermore, more frequent autumn precipitation as a direct result of incomplete or late sea ice cover during the late Miocene may have contributed to thicker snow cover during winter, insulating the ground and hampering the formation of permafrost” – is it not more likely that this fell as rainfall?

Our reply:

The reviewer is correct regarding the autumn, in case of MAAT ranging between +6.6°C and +11.1°C, it most probably fell as rain. With the winter temperatures ranging from +1°C to -3°C

(Popova et al 2012), the ground would be frozen only briefly and to shallow depth only, with no permafrost at all.

We wrote: “These lines of evidence imply a temperate climate and an ice-free Arctic Ocean (at least during summer) in the Tortonian (ref 27) (Fig. 2). Under such climate conditions, only brief, if any, ground freezing in mid-winter would occur, not hampering speleothem growth that was enhanced by high Miocene mean-annual precipitation of 800-900 mm (ref 26).” (Lines 94-97).

The reviewer:

Line 93-95 – “Based on the range of reconstructed temperatures, we estimate the Tortonian northern Siberia featured a MAAT in the range of -3 to +16 °C, equivalent to an increase of 9 to 28°C from present-day temperatures.”

-This statement is a bit strong, given that the temperature estimates are from other studies. It could be more nuanced to say something like “Based on the range of reconstructed temperatures (refs), Tortonian northern Siberia MAAT is estimated in the range of -3 to +16 °C, equivalent to an increase of 9 to 28°C from present-day temperatures.”

Our reply:

We changed this statement as we wrote above, lines 94-97.

The reviewer:

Line 101 – ‘any speleothem formation implies a MAAT of >-5°C, ‘

Please expand and explain further. If the cave air temperature is representative of MAAT, then at 0 to -5°C, cryogenic cave carbonates would form. However, as the authors state above, increased snowfall may insulate the ground – in this case, the cave air temperature will be decoupled from MAAT. This also excludes issues of cave geometry, which I appreciate the authors cannot possibly know with the relics that they have today.

Our reply:

Vadose speleothems, stalactites, stalagmites and flowstone, form if permafrost is discontinuous and enables water seepage into the cave. The MAAT at which permafrost is discontinuous is 0°C to -6°C (ref 23). Compared to present-day MAAT of -12.3°C, this defines the minimal increase in MAAT of ~6°C required to allow the onset of vadose speleothem deposition. However, the paleotemperatures we received from clumped isotope analyses of these speleothems place the Tortonian MAAT to +6.6°C - +11.1°C, which minimizes speculation about the minimal MAAT/MAGT required to cause speleothem growth. Therefore this part of the discussion was mostly removed from the manuscript, with only brief mentioning of:

“The minimum MAAT range that leads to discontinuous permafrost and enables water penetration into the caves is 0°C to -6°C (ref 23), however, clumped isotope analyses performed on four Taba-Ba’astakh speleothems show paleotemperatures of +6.6°C - +11.1°C (ref 4).” (lines 68-70).

The reviewer:

Line 107 - '9-20°C of warming'

-Perhaps specify that this is regional (not global) warming just for clarity

Our reply:

We removed "9-20°C of warming", writing instead: "Considering today's MAAT of -12.3°C at Taba-Ba'astakh, we chose an increase of 18.9°C above present in polar regions of the Northern Hemisphere, to MAAT of +6.6°C obtained from clumped isotope measurement of Taba-Ba'astakh speleothems (ref 4). This is our conservative estimate of warming during the Tortonian (Fig. 3-A and 3-B, Dataset-2, Methods)." (Lines 103-106)

The reviewer:

Line 108 onwards – discussion related to permafrost thaw and release of carbon. This is not really my expertise so I cannot comment on the details. It is not clear to me though, why Umbo et al. are already updating this Vaks et al paper

Our reply:

In the revised version of this manuscript we cite the MAAT of +6.6°C - +11.1°C reconstructed by Umbo et al. 2024 (ref 4), and our corrected GIS analysis of Tortonian extent of permafrost is based on these paleotemperatures. The Umbo et al. 2024 paper was not cited in our study because it was not submitted at the time. This is also the reason why Umbo et al. used 'updated' information. This has been rectified.

The reviewer:

Umbo et al 'Using our new temperature reconstructions, we estimate total potential permafrost derived carbon emissions given future warming similar to that reconstructed for the Tortonian and update the previous calculation from Vaks et al. (in review)' Lines 491-492. Furthermore, figure 3 in this paper is almost identical to figure 7 in Umbo et al. though admittedly with different temperature estimates.

Our reply:

We revised both papers, this one and the paper of Umbo et al. (2024), ref 4, and now each paper focusses on completely different aspects and serving different purposes. Umbo et al. (2024) uses Taba-Ba'astakh speleothems for the reconstruction of Tortonian MAAT using clumped isotope analysis and proxy information to gain insights into the Miocene palaeoenvironment at Taba-Ba'astakh. Our paper reconstructs the chronology of 14 Taba-Ba'astakh speleothems, and then uses the MAAT estimate of Umbo et al. (2024) for the reconstruction of Tortonian permafrost extent in the Northern Hemisphere using GIS model results.

The reviewer:

References – please check as they are not all formatted the same, differences in style of dois for example. References 7 and 31 have doi written twice.

Our reply:

Four references with this type of error were corrected.

Co-reviewer

Specific comments

The reviewer:

Abstract (Lines 4-8): The opening lines would benefit from more precise language than "climate change is happening faster in Siberia than globally". What kind of climate change? How much faster? How much permafrost thaw qualifies as 'substantial'?

Our reply:

Thank you, we agree and have rephrased this to read:

"Arctic warming is happening at nearly four times of the global average rate (ref 1). Long-term trends of permafrost dynamics cannot be estimated directly from monitoring of present-day thaw processes, requiring paleoclimate-proxy information. (Lines 5-7)

The reviewer:

More importantly, if the goal is to assess the potential for Siberian permafrost thaw with Anthropogenic global warming (AGW) over the next century, the Late Miocene is unlikely to be very helpful, for reasons mostly given later in the manuscript (e.g. lack of Greenland ice sheet, which significantly cools the modern Arctic; partially open Panamanian seaway, altering the NAC/Gulf Stream dynamics; closed Bering Strait; expanded Arctic coastlines and much lower sea-ice limits, enhancing precipitation). For instance, even though CO₂ concentrations were lower than at present, the abrupt post-YD warming during the Early Holocene constitutes a problematic analog to Siberian permafrost degradation from AGW, due to several important differences in the boundary conditions. For example, see discussion and references in Li et al. (2021; <https://www.nature.com/articles/s43247-021-00238-z>).

Our reply:

In the late Miocene the global Mean Annual Surface Temperature (MAST) was 4.5°C higher than present. Although the causes to this warmth were not anthropogenic, this still shows what may happen with today's permafrost in one of the most extreme warming scenarios that will bring the MAST to such high values.

The reviewer:

In short, documenting the Miocene permafrost limits is a major scientific advancement and will greatly elucidate high-latitude climate sensitivities. However, it provides limited information on how much permafrost retreat could happen in response to comparable changes in global MAT today. Under no circumstances does Miocene (vs. Holocene) warming in the Arctic, highly

amplified by the paucity of NH ice sheets, describe a reasonable expectation for near-term anthropogenic warming. Hence, I consider the end goal of calculating carbon budgets from permafrost thaw under Miocene climate to be somewhat questionable in 21st-century (or longer) forecasts.

Our reply:

The late-Miocene level warming scenario gives insights into what *may* occur in the Arctic under extreme global warming of 4.5°C above present. It is not aiming to show what will *definitely* happen till the end of 21st century, but provides an estimate of the area where permafrost will become vulnerable to thaw, as well as the amount of carbon that can be potentially released under such scenario.

The reviewer:

From Popova et al., 2012:

Late Miocene MAAT is near ~12-14°C for coastal northern/eastern Siberia, with CMT around -4 to 0°C, which corresponds to warming of ~26°C (MAAT) and ~30°C (CMT) relative to today (which is already warmer than PI). Using the (slightly lower) Omoloy River data, there is a range of ~20-29°C (MAAT) and ~26°C (CMT) higher than today.

- Why reference these data and then provide such a vague warming estimate of >20°C, especially when it follows such a specific lower range of 9°C? Please consider taking them at face value and provide a specific range, or simply use the upper limit of their reconstruction in all instances (e.g. "Miocene warming along the Arctic coast up to 29°C higher than today").

Our reply:

In the revised manuscript we cite the MAAT of +6.6°C - +11.1°C as reconstructed by Umbo et al. (2024, ref 4), which are based on clumped isotope analyses on several of the same speleothems dated in this study. This means that the Tortonian MAAT at Taba-Ba'astakh was 18.9°C - 23.4°C higher than today.

The reviewer:

- Late Miocene precipitation (800-900 mm MAP) is estimated to be slightly higher than the Pliocene (700-800 mm MAP). As far as I can tell, we aren't provided any indication of modern MAP at the cave site, with which to compare these earlier values. But these data are highly important to assessing the relevance of MAAT changes and the expected ecological evolution of a post-permafrost landscape.

Our reply:

We entered the data on modern MAP at the research site and it is ~300 mm, see line 43 in the manuscript.

The reviewer:

Line 29-30: "Mean annual ground warming above 0°C..." Is this supposed to read mean-annual ground temperature (MAGT)? If so, please clarify it in the text. Either way, the discussion would benefit from considering the various thresholds for permafrost cover and the relationship between permafrost, ground temperature, and surface air temperature.

Our reply:

We wrote in lines 25-26: "An increase of mean annual ground temperature (MAGT) above 0°C can..."

The reviewer:

For example, permafrost-stored soil carbon is already susceptible to mobilization at the continuous-discontinuous boundary (around $-1.7 \pm 0.5^\circ\text{C}$; Obu et al., 2019), which corresponds (in northern Siberia) to MAAT between -9°C and -6°C (Gruber et al., 2012). The modern study site is already within 5°C of this threshold.

Our reply:

In the revised manuscript we cite the paleo-MAAT of $+6.6^\circ\text{C}$ - $+11.1^\circ\text{C}$ obtained from the same speleothems. This temperature range is well above 0°C , therefore MAGT below 0°C can be excluded under these conditions.

We wrote in lines 63-65: "The presence of multiple Middle-Late Tortonian speleothems at the Taba-Ba'astakh site shows that at 8.68 ± 0.09 Ma this present-day tundra region experienced a more temperate climate, with MAGT above 0°C and with permafrost-free conditions."

The reviewer:

Line 31-32: What is the source of this "considerable uncertainty"? Shouldn't sustained MAGT $>0^\circ\text{C}$ do the trick? If so, calculate the difference between present ground temperatures and 0°C , and that's how much warming is needed. Beyond that, it's only a matter of time (which does have some uncertainty, but as far as I can tell, thaw rates are not addressed in this study).

Our reply:

Given the fact that the magnitude of warming in polar regions is much higher than the warming of Mean Earth Surface Temperatures (MEST), uncertainty exists largely in determining which increase in MEST is required to cause the warming of polar regions in order to bring the MAGT there above 0°C .

We changed this sentence to:

"Considerable uncertainty still exists, however, about the degree of global warming required to completely thaw the near-surface permafrost presently located in the coldest Northern Hemisphere regions such as northern Siberia." (Lines 27-29).

The reviewer:

Line 42-52: The site description is good, but it need not be repeated verbatim in the SI and it lacks two vital components: 1) What is the monthly/seasonal precipitation, locally? 2) What are the ground temperatures near the site? Since the focus is on permafrost extent and

degradation, ground temperature is a more important component than air.

Our reply:

“The region is characterized by a tundra climate, with average temperatures ranging between -31.6°C in January and +9.3°C in July (MAAT of -12.3°C, Fig. 1) (ref 17), by annual precipitation of ~300 mm (ref 18), by continuous permafrost with thicknesses of 300-500 meters, and mean annual ground temperatures (MAGT) of -9°C - -11°C (ref 2) (see also Methods).”.

The reviewer:

U-Pb dating: I congratulate the authors on this stunning geochronological result. Can you clarify in the text, was any attempt made to constrain the range of speleothem-recorded growth? How thin of a snapshot are we talking? I appreciate this may not have been plausible with the sample selection (especially if they're nearly coeval), but if that's the case, there should be mention of how much time feasibly could be recorded by these speleothems.

Our reply:

Yes, we did consider the potential age range, but unfortunately it was not really feasible to determine a growth duration in a particularly useful way. To do so would require a majority (i.e. at least a representative subset) of the samples to yield sufficiently precise ^{235}U - ^{207}Pb model ages that either real age variations were resolvable, or it is possible to say the material is of the same age within a useful level of precision; only 3 of 14 speleothems are really useful in this regard. Many of the other samples are unradiogenic and yield either no, or only imprecise model ages, such that there is limited ability to usefully test for age differences between them. Moreover, because of the hyperbolic inflation of the age uncertainty with common Pb content, any age range limit determined for a wider subset of samples (beyond the 3 most radiogenic samples) basically ends up being an artefact of the common Pb correction uncertainty that depends on the maximum $^{208}\text{Pb}/^{207}\text{Pb}$ of the subset of data considered.

We added the detailed discussion of this topic in the Methods section, as well as a new Figure 13 for explanation:

“In terms of the potential growth interval collectively represented by the Taba-Ba'astakh samples, all but one of the 66 analyses fall on the isochron in $^{208}\text{Pb}/^{207}\text{Pb}$ - $^{235}\text{U}/^{207}\text{Pb}$ space (Fig. 12) such that the samples are mostly unresolvable from having formed instantaneously at the c. 8.7 Ma isochron age (Fig. 13). However, the resolution with which potential age variations could be detected varies greatly between the samples depending on their radiogenicity (Fig. 13). For many, the large model age uncertainty arising from the common Pb correction negates using them in any meaningful way in constraining the growth interval recorded (Fig. 13). The most radiogenic analyses suggest a growth interval no longer than a few hundred-thousand of years, but represents only 3 of 14 speleothems (see below). With only a minority of the samples being adequately datable, little else can be said with confidence about the growth interval.” (Lines 281-290).

We also added the following sentences in the main text:

“The total growth interval represented by the samples is ambiguous as many of them are unradiogenic, yielding imprecise model ^{235}U - ^{207}Pb ages. However, based on the most radiogenic material, a growth interval of no more than a few hundreds of thousands of years around the isochron age is most likely.” (Lines 56-59).

The reviewer:

Line 101: "and the fact that any speleothem formation implies a MAAT of $>-5^{\circ}\text{C}$..."

Why -5°C ? Because of conditions around Botovskaya? Either way, this is an oversimplification of cryospheric processes and thresholds. This sentence could be revised to something like:

"Considering today's MAGT of $-XX^{\circ}\text{C}$ at Taba-Ba'astakh and the fact that any speleothem formation implies a MAGT of at least -2°C (i.e. discontinuous permafrost) but more likely $>0^{\circ}\text{C}$..."

Our reply:

By -5°C and higher we meant the MAAT required for discontinuous permafrost. However, considering the Umbo et al. (2024) MAAT estimate of $+6.6^{\circ}\text{C}$ as a minimum, then it is irrelevant. We rephrased the sentence as follows (Lines 103-106):

“Considering today's MAAT of -12.3°C at Taba-Ba'astakh, we chose an increase of 18.9°C above present in polar regions of the Northern Hemisphere, to MAAT of $+6.6^{\circ}\text{C}$ obtained from clumped isotope measurement of Taba-Ba'astakh speleothems (ref 4). This is our conservative estimate of warming during the Tortonian (Fig. 3-A and 3-B, Dataset-2, Methods).”

The reviewer:

By relating the discussion to MAAT, the complex relationship between ground and air temperatures is missing, which is important in regions like Siberia with high-amplitude seasonality and substantial winter snowpack (though I appreciate that the insulating effect of snow is briefly mentioned):

Smith, M.W., and D.W. Riseborough 1996. Permafrost Monitoring and Detection of Climate Change, Permafrost and Periglacial Processes, 7, 301–309.

Wright et al., 2003, Regional-scale permafrost mapping using the TTOP ground temperature model

Gruber et al., 2012 - <https://doi.org/10.5194/tc-6-221-2012>

Obu et al., 2019 - <https://doi.org/10.1016/j.earscirev.2019.04.023>

Peng et al., 2024 - <https://doi.org/10.1088/1748-9326/ad30a5>

The Botovskaya Cave region is not necessarily informative of the air-ground insulating effect along the Arctic coastline, because the determining factors are the relative proportion of freezing and thawing degree days (a function of seasonal amplitude), their respective conductivities, and snow-cover days. Hence that air-ground temperature difference of $\sim 5^{\circ}\text{C}$ near Botovskaya climbs to $\sim 9^{\circ}\text{C}$ near the Taba-Ba'astakh site. Without specific constraints on changes in winter precipitation and summer-winter amplitude during the Miocene, it is difficult to estimate the magnitude shift in MAAT required to induce permafrost thaw at the modern Arctic coastline (i.e. a $+9^{\circ}\text{C}$ change in MAAT does not necessarily equate to a $+9^{\circ}\text{C}$ change in ground temperature). Please consider incorporating this in the discussion more comprehensively or

limit the quantitative estimates to ground-temperature changes as a combined, yet uncertain, function of surface temperature plus winter precipitation. I think the latter almost achieved already but could be improved a little.

Our reply:

We mentioned Botovskaya only as an active cave located in today's discontinuous permafrost zone, similar to the conditions in which Taba-Ba'astakh could be if the temperatures were only 4.3°C to 12.3°C warmer, placing it in discontinuous permafrost zone. However, Umbo et al. (2024), ref 4, shows that this wasn't the case and temperatures were at least 18.9°C higher, placing the research area in the non-permafrost zone. Therefore, we removed from the manuscript the discussion about the minimal MAAT warming of +9°C, that uses Botovskaya cave as an example.

The reviewer:

Line 114-118: "Estimates by Schuur et al. suggest that 5-15% of this surficial soil organic carbon would be vulnerable to be released as greenhouse gases within decades in the short-term, while longer-term estimates for soil carbon release are poorly constrained. This means that 30-85 PgC will be released into the atmosphere for the conservative estimate of uniform warming of 9°C across Siberia (Fig. 3-C)."

Our reply:

We have included the most recent publication by Schuur et al. (2022, <https://doi.org/10.1146/annurev-environ-012220-011847>) in the list of references (ref 6) to the sentence here – this reference is a high-level summary of the carbon cycle feedbacks of permafrost thaw in the Arctic, and it includes many of the experts on this topic including Turetsky. Our manuscript does not focus on the many complex feedbacks and uncertainties involving peatland formation – we simply use a higher level estimate for the percentage of carbon that would be released on a short term basis. However, we did include a new sentence that points to the complexities of calculating carbon cycling:

"Estimating permafrost carbon release and uptake is uncertain because it depends on heterogeneous landscape factors such as ice content, abundance and distribution of lakes, wetlands, and vegetation composition that shift as the landscape evolves (ref 30)." lines 118-120), as well as "Schuur et al.(ref 6, 7) **(and ref within)** suggest that 5-15% of this surficial soil organic carbon would be vulnerable to be released as greenhouse gases within decades in the short-term, while longer-term estimates for soil carbon release are poorly constrained." (Lines 120-122)

The reviewer:

In my opinion, the application of modeling studies by Schuur et al. needs to be refined, and much of the vast literature since 2015 on this topic has been left out. First, I am not convinced that the cited range of Siberian warming (9-29°C) is realistic for global MAAT forecasts up to +4.5°C, unless you can also remove the Greenland ice sheet in a matter of decades this century.

Our reply:

As we mentioned above, Miocene global warming of MEST by 4.5°C can be used to estimate the potential degree of polar warming in extreme scenarios. The threat of rapid melting of large volumes of the Greenland ice sheet just underscores the importance of our results.

The reviewer:

Second, the estimates for carbon mobilization are based on limited data, hence Schuur et al. caution that several uncertainties exist. These processes have been extensively investigated since Schuur et al. 2015 with updated datasets (refs below), including the find that relatively little of the carbon release during permafrost and peatland warming is from ancient carbon stores, at least during the Early Holocene. Consequently, the manuscript would benefit from refining its discussion to include some of the vast literature post-2015 that has not been cited. Otherwise, it risks oversimplifying cryospheric processes in terms of carbon cycling. Finally, carbon cycling amid abrupt permafrost degradation is complicated by the specific ecological response, particularly with regard to peatland formation:

Turetsky et al., 2020, <https://www.nature.com/articles/s41561-019-0526-0> (as a start; almost any recent paper by Merritt Turetsky will be highly relevant)

Smith et al., 2004, Siberian peatlands a net carbon sink and global methane source since the Early Holocene

MacDonald et al., 2006, Rapid early development of circumarctic peatlands and atmospheric CH₄ and CO₂ variations.

Alexandrov et al., 2016, The influence of climate on peatland extent in Western Siberia since the Last Glacial Maximum

In short, moisture availability and rates of future warming are important factors in constraining carbon mobilization to atmospheric reservoirs versus terrestrial uptake through peatland expansion. Providing a simple fraction of carbon stored within given isotherms is again misleading, because it belies the complex but well studied dynamics of carbon cycling and storage in permafrost and peatlands. Under what conditions would carbon mobilization to the atmosphere be amplified or mitigated during permafrost thaw in the high Arctic of Siberia?

Our reply:

We agree that carbon mobilization is a highly complex process, and we don't attempt to dive into the dynamics relevant to each ecological system. The warming rate is given by RCP 8.5, where we cite Schuur et al. (2015, 2022), who estimate a release "between 5 and 15% of the permafrost carbon pool over decades to centuries under business-as-usual warming scenarios [Representative Concentration Pathway (RCP) 8.5], rather than as a catastrophic pulse release on the scale of a few years." Our estimate of carbon stored up to a given isotherm is useful because it accounts for the variability of carbon stocks in the areas that are likely to thaw first, as opposed to the carbon retained in isotherms that are much cooler. We are not attempting to answer the detailed questions of amplification or mitigation of carbon mobilization – instead we are applying a generalized estimate based on the recent literature.

The reviewer:

I hope the authors can develop a more nuanced discussion of these dynamics, because this speleothem-based record will provide an important contribution to the field and to a time and region that is scarcely represented by terrestrial proxy data.

Our reply:

We addressed the comments of reviewer #2, and made appropriate changes in the manuscript accordingly.

Reviewer #3 (Remarks to the Author):

dear authors,

As requested, I read and reviewed the manuscript titled “Arctic speleothems reveal Late Miocene permafrost-free Northern Hemisphere” by A. Vaks and co-workers submitted for publication in Nature Communications. In their study, the authors present precise ages of speleothems found in a cave system in northern Siberia and argue that the youngest age of these speleothems represents the last time this region was free of permafrost. They use this result together with observations in other caves in the region to determine the degree of warming that is needed to melt the permafrost in the Arctic region. They then use this threshold to inform models of soil organic carbon stored in Siberia under various global warming scenarios to estimate how much carbon could be released from melting permafrost in the future. This is a well-written manuscript with a clear and important message with relevance for the global climate crisis: The melting of permafrost in future scenarios is one of the most important and most uncertain response mechanisms that could have an accelerating effect on global warming. Therefore, the effort this study makes to better understand the sensitivity to Arctic permafrost systems to warming is of high societal and scientific importance. I find the arguments put forward by the authors to support that the soil above the northern Siberian site under study was at least discontinuously frozen convincing. The way the authors combine observational evidence with modelling, especially embracing the wide array of possible paleotemperatures for the site during the Tortonian, is thoughtful. While I am no expert in interpreting the palynological/vegetation evidence, I find the lines of reasoning put forward by the authors plausible and I am convinced that their conclusions about the sensitivity of the permafrost-stored carbon pool to warming are sound within their given uncertainty. Two questions occurred to me while reading this work, which I think the authors might be able to answer by expanding upon these subjects in the text. Beyond these more general comments, I detail some additional minor comments which, while not impactful regarding the scientific quality of the study, might help the authors improve their manuscript to more clearly convey its main points. I trust the authors will be able to address these concerns and make the manuscript acceptable for publication after minor revisions.

Seasonal variability

The reviewer:

Firstly, given the large temperature seasonality in the region today (see line 45). I wonder whether the presence of speleothem formation at the site in the Tortonian proves that the soil in region was ice-free at this time. Is it not possible that speleothem formation took place during part of the year (e.g. only in summer) and that the soil was completely frozen in the winter? It is not clear from the text or the supplement for how much of the annual temperature cycle the air temperature needs to be above freezing for speleothem formation to occur. It is also not clear whether seasonal temperature variability is taken into account in the soil carbon modelling exercise. Therefore, it remains unclear (to me) how partial, seasonal, or discontinuous freezing of the soil affects the amount of carbon that will be released when the threshold annual air temperature is reached at the soil location.

Our reply:

The definition of the permafrost is: “Permanently frozen ground... .. under thermal conditions where temperatures below 0°C have persisted for at least two consecutive winters and intervening summer” (Oxford Dictionary of Earth Sciences’ 2008, Third Edition, M. Allaby Ed.). Therefore there is no seasonal unfreezing of permafrost, although during summer an “active layer” of soil and rock (several cm to 2 m thick) will thaw. However, if the permafrost is continuous, the frozen ground below the active layer prevents any water seeping into the caves that remain frozen in permafrost year-round. If water does find its way into the caves, the permafrost is not continuous. This may happen when the MAAT reaches a level between 0°C and -8°C, which at Taba-Bastaakh requires warming of at least 4.3°C to 12.3°C compared to present. According to Umbo et al. (2024), ref 4, the minimum MAAT was much higher, +6.6 °C - +11.1°C, which means that permafrost would be non-existent at the research site at the time of speleothem formation.

The reviewer:

If the observed evidence of speleothem growth can be achieved with a seasonal thawing of the permafrost, I think the title of the manuscript (mentioning a “permafrost-free Northern Hemisphere”) is too strong. In that case I would suggest the authors refer in their title to the large amount of carbon that can be released from the Arctic permafrost rather than making statements about the complete disappearance of permafrost in the region, which I believe is not fully supported by the data. The former is, in my opinion, a more important outcome of the study than the (perhaps partly semantic) discussion of whether or not the whole Arctic region was permafrost free.

Our reply:

The paleotemperature evidence provided above (Umbo et al. 2024, ref 4) shows that, indeed permafrost-free conditions prevailed at the time. GIS analyses shows that it is likely that the entire Northern Hemisphere should be mostly permafrost-free in case of 18.9°C warming, and completely permafrost-free if the warming at Taba-Ba’astakh would be 20°C or higher.

The Tortonian as an analogue for future climate

The reviewer:

The Tortonian features a mean annual global temperature of 4.5 degrees above the pre-industrial temperatures. This is a relatively high temperature scenario when compared to the IPCC scenarios which are currently most in agreement with our global policies and actions. In the introduction, I get the feeling that the authors argue that 2-5 degrees of warming is realistic in the near future (they do not clearly specify exactly when these temperatures could be reached). In my opinion, the manuscript would benefit from a clearer statement on our current global warming scenario and how the Tortonian climate compares to it. I am not arguing that, because the Tortonian was warmer than the climate we are likely to experience in the year 2100, the Tortonian climate is not an interesting case study to better understand the response of permafrost to warming. However, it would be useful for the less-initiated reader to see the Tortonian climate as representative for (if not completely analogous to) a high-warming climate scenario to place the results of this study in that context.

Our reply:

We do not attempt to make a forecast of the scale of warming that will take place until 2100, or 2300. Instead, we use the Tortonian warm climate in the Arctic to show what can be the final outcome of the most extreme warming scenario.

We wrote in lines 132-134:

“Whilst such boundary conditions differed to those of the present-day, a permafrost-free Tortonian Siberia still represents a scenario with $\sim 4.5^{\circ}\text{C}$ warming of global MAAT, closest to the most extreme warming scenarios proposed by the IPCC (refs 9, 10).”

The reviewer:

Minor comments

- Line 101: The authors state that speleothem formation requires an MAAT above -5°C , but it is not clear what this minimum temperature is based on. Please add some references for this and explain briefly how this threshold is determined.

Our reply:

The $>-5^{\circ}\text{C}$ MAAT is usually required to cause the permafrost to become discontinuous and enable speleothem deposition. However, because of Tortonian paleo-MAAT of $+6.6^{\circ}\text{C}$ - $+11.1^{\circ}\text{C}$ determined using clumped isotope analysis on the same speleothems (Umbo et al 2024), ref 4, the warming that occurred in the Tortonian was much more intensive than the conservative scenario of only causing the permafrost to become discontinuous. Therefore, we changed this sentence to the following: This sentence was changed to “Considering today’s MAAT of -12.3°C at Taba-Ba’astakh, we chose an increase of 18.9°C above present in polar regions of the Northern Hemisphere, to MAAT of $+6.6^{\circ}\text{C}$ obtained from clumped isotope measurement of Taba-Ba’astakh speleothems (ref 4). This is our conservative estimate of warming during the Tortonian (Fig. 3-A and 3-B, Dataset-2, Methods). (Lines 103-106).

”.

The reviewer:

- Figure 1: The black numbers next to the black stars indicating other speleothem sites are hard to read due to the color contrast. Perhaps the authors can slightly change the background of

these numbers to make them stand out a bit more. Otherwise this figure is very clear and conveys a lot of important information in a smart way.

Our reply:

We changed the black stars to yellow ones, now it looks better. We also removed Botovskaya Cave from all the maps, because it is not cited in the revised manuscript.

The reviewer:

- Figure 2: The diamond indicating the isochron age looks more red than purple to me and is very similar in color to the red curve indicating the western Arctic temperature evolution. Perhaps the authors can make it stand out a bit more by adding more blue to the purple.

Our reply:

We changed the color to blue.

The reviewer:

In addition, in the text the difference between the two dates and their uncertainties (isochron vs 19 most radiogenic samples) is not clearly explained. In the Methods section (line 110-112), the authors argue that the isochron age to be the most precise age determination. I would suggest the authors either use this ages as representative for the most recent speleothem formation or explain in the main text why two different age estimates are presented to prevent confusion to the more general reader.

Our reply:

The two age determinations are given in order to show both the isochron age (the short interval of 8.68 ± 0.09 Ma), as well as the most likely duration of the speleothem deposition period represented by the longer interval of (8.7 ± 0.4) Ma). As we explain to reviewer 2, our ability to constrain the growth period is limited because it would require 66 individual ^{235}U - ^{207}Pb ages. However only 19 samples are radiogenic enough to obtain relatively accurate individual ages, clustering in 8.7 ± 0.4 Ma. Other samples have too high proportion of non-radiogenic lead, which increases the age uncertainty on their individual ages (although they still cluster around ~ 8.7 Ma). In terms of $^{208}\text{Pb}/^{207}\text{Pb} - ^{235}\text{U}/^{207}\text{Pb}$ isotope space, most of the data are consistent with the c. 8.7 Ma isochron and hence with this age, within analytical precision of 0.09 Ma (95% confidence level), which is the shorter interval we show. We added the detailed discussion of this topic in the Methods section (Lines 278-311, Methods). We added in main text “Results” section: “The total growth interval represented by the samples is ambiguous as many of them are unradiogenic, yielding imprecise model ^{235}U - ^{207}Pb ages. However, based on the most radiogenic material, a growth interval of no more than a few hundreds of thousands of years around the isochron age is most likely.” (lines 56-59).

References

Climate Action Tracker. 2100 Warming Projections: Emissions and expected warming based on pledges and current policies. <https://climateactiontracker.org/global/temperatures/> (2023).

Reviewer #4 (Remarks to the Author):

The reviewer:

Our reply:

All comments are addressed above.

Dear Editor,

Please find below the point-to-point rebuttal letter, where the reviewers answers from the second revision round are marked in red, and our answers are marked in green. Please see the reviewers comments that required answers followed by our replies on pages 6-13.

REVIEWERS' COMMENTS

Reviewer #1 (Remarks to the Author):

I reviewed an earlier version of this manuscript and offered only minor comments. These have been adequately addressed. **In my opinion, the manuscript is acceptable for publication.**

Reviewer #2 (Remarks to the Author):

Thank you for the opportunity to read the revised version of Vaks et al., 'Arctic speleothems reveal Late Miocene permafrost-free Northern Hemisphere'. We consider this to be a valuable study of great importance to furthering our understanding of High Latitude climate and permafrost extent. **We look forward to seeing it published.**

'Established reviewer'

Specific comments

Line 10: the authors state that they are using cave carbonates to determine when northern Siberia was last permafrost-free. I find this to be quite a sweeping statement that could be more nuanced. For instance, there are many reasons why the region may be permafrost-free but speleothems did not form – hence the presence of speleothems may not necessarily date to the last time it was permafrost-free. The number of samples analysed is also relatively small, hence there may be an element of sampling bias involved. Finally, northern Siberia is very large, surely the results from this study are only regional in extent and do not cover the whole of northern Siberia?

- In the response, the reviewers outline a number of valid reasons to address this issue. The abstract has been updated and is more nuanced though I don't see the specific arguments put forward in the rebuttal as having made it into the manuscript (which would be a great help to readers who are not experts in speleothem growth in High Arctic environments). Despite this, **I agree with the scientific reasoning and don't have issues with it published as it is.**

Line 13 – I suggest this is just 'late', and delete 'middle'

- **The authors have addressed this.**

Line 16 – 'while MAAT at the Siberian Arctic coast were 9oC to >20oC above present' – if I check the reference that is given (Popova et al., 2012) then it gives MAAT of 9oC but

the >20°C relates to the warm months only (not MAAT). The highest MAAT from this region seems to be +16 °C from the Omoloy river.

- The authors have addressed this.

Lines 17-19 – ‘Our findings provide direct observational evidence that warming to temperatures similar to those of the Tortonian period would lead to extensive permafrost thaw along the Siberian Arctic coast’.

I agree there is no doubt that warming to such temperatures would lead to extensive permafrost thaw, however, the statement implies that the temperatures that are being considered here are the 9°C and >20°C from the previous sentence, which comes from another study. The issue is that: (1) the Vaks manuscript doesn’t seem to provide direct observational evidence related to quantitative temperature reconstructions (because it comes from another study), and (2) I am not sure why the authors are not citing their own work on quantitative temperature reconstructions that is in review elsewhere (Umbo et al.).

- The authors have addressed this.

Line 36 – if I understand correctly, the caves are from one location, hence ‘northern Siberian regions’ may be a little exaggerated

- I am satisfied with this response and thank the authors for explaining the thought process.

Line 37-38 - “Speleothems found in today’s permafrost regions attest to warmer periods when continuous permafrost was absent to allow water seepage into caves“ – it is certainly possible for water to enter caves even when they are in permafrost if there is enough latent heat in the flow and a route for it to take such as an aven or fissure. Water is also required for cryogenic cave carbonates to form, which are a form of speleothem. However, common speleothems such as stalactites, stalagmites and flowstones indeed require a temperature above 0°C so that the water does not freeze. Perhaps this sentence can be revised e.g., “Common speleothems found in today’s permafrost regions attest to warmer periods when continuous permafrost was absent and cave temperatures above freezing”

- The authors have addressed this.

Line 61-63 – ‘The geographical position of eastern Siberia did not change significantly over the last 80 Ma, suggesting that plate-movement-related latitude changes did not affect climate in the region.’ – A comment on uplift would be appreciated so that we know whether the caves are expected to be in the same position as the time of speleothem deposition.

- It is a little buried and could be more upfront, but the authors have addressed this.

Lines 80-85 – this comes back to my earlier comment. Whilst this discussion is relevant and very interesting, why do the authors not refer to their own work, which firmly places the temperature estimate between +6.6 and +11.1 °C.

- The authors have addressed this.

Line 91-93 - “Furthermore, more frequent autumn precipitation as a direct result of incomplete or late sea ice cover during the late Miocene²⁵ may have contributed to thicker snow cover during winter, insulating the ground and hampering the formation of permafrost” – is it not more likely that this fell as rainfall?

- The authors have addressed this.

Line 93-95 – “Based on the range of reconstructed temperatures, we estimate the Tortonian northern Siberia featured a MAAT in the range of -3 to +16 °C, equivalent to an increase of 9 to 28°C from present-day temperatures.”

-This statement is a bit strong, given that the temperature estimates are from other studies. It could be more nuanced to say something like “Based on the range of reconstructed temperatures (refs), Tortonian northern Siberia MAAT is estimated in the range of -3 to +16 °C, equivalent to an increase of 9 to 28°C from present-day temperatures.”

-The authors have addressed this.

Line 101 – ‘any speleothem formation implies a MAAT of >-5°C, ‘

Please expand and explain further. If the cave air temperature is representative of MAAT, then at 0 to -5°C , cryogenic cave carbonates would form. However, as the authors state above, increased snowfall may insulate the ground – in this case, the cave air temperature will be decoupled from MAAT. This also excludes issues of cave geometry, which I appreciate the authors cannot possibly know with the relics that they have today.

- The authors have addressed this.

Line 107 - ‘9-20°C of warming’. Perhaps specify that this is regional (not global) warming just for clarity

- The authors have addressed this.

Line 108 onwards – discussion related to permafrost thaw and release of carbon. This is not really my expertise so I cannot comment on the details. It is not clear to me though, why Umbo et al. are already ‘ updating’ this Vaks et al paper

Umbo et al ‘Using our new temperature reconstructions, we estimate total potential permafrost derived carbon emissions given future warming similar to that reconstructed for the Tortonian and update the previous calculation from Vaks et al. (in review)’ Lines

491-492. Furthermore, figure 3 in this paper is almost identical to figure 7 in Umbo et al. though admittedly with different temperature estimates.

- The authors have addressed this.

References – please check as they are not all formatted the same, differences in style of dois for example.

References 7 and 31 have doi written twice

- The authors have addressed this.

Co-reviewer

Specific comments

Abstract (Lines 4-8): The opening lines would benefit from more precise language than "climate change is happening faster in Siberia than globally". What kind of climate change? How much faster? How much permafrost thaw qualifies as 'substantial'?

- The authors have addressed this comment.

More importantly, if the goal is to assess the potential for Siberian permafrost thaw with Anthropogenic global warming (AGW) over the next century, the Late Miocene is unlikely to be very helpful, for reasons mostly given later in the manuscript (e.g. lack of Greenland ice sheet, which significantly cools the modern Arctic; partially open Panamanian seaway, altering the NAC/Gulf Stream dynamics; closed Bering Strait; expanded Arctic coastlines and much lower sea-ice limits, enhancing precipitation). For instance, even though CO₂ concentrations were lower than at present, the abrupt post-YD warming during the Early Holocene constitutes a problematic analog to Siberian permafrost degradation from AGW, due to several important differences in the boundary conditions. For example, see discussion and references in Li et al. (2021; <https://www.nature.com/articles/s43247-021-00238-z>).

In short, documenting the Miocene permafrost limits is a major scientific advancement and will greatly elucidate high-latitude climate sensitivities. However, it provides limited information on how much permafrost retreat could happen in response to comparable changes in global MAT today. Under no circumstances does Miocene (vs. Holocene) warming in the Arctic, highly amplified by the paucity of NH ice sheets, describe a reasonable expectation for near-term anthropogenic warming. Hence, I consider the end goal of calculating carbon budgets from permafrost thaw under Miocene climate to be somewhat questionable in 21st-century (or longer) forecasts.

- The authors have sufficiently addressed my concerns in the updated text. I appreciate the clarifications on the difference in boundary conditions.

From Popova et al., 2012:

Late Miocene MAAT is near ~12-14°C for coastal northern/eastern Siberia, with CMT around -4 to 0°C, which corresponds to warming of ~26°C (MAAT) and ~30°C (CMT) relative to today (which is already warmer than PI). Using the (slightly lower) Omoloy River data, there is a range of ~20-29°C (MAAT) and ~26°C (CMT) higher than today.

- Why reference these data and then provide such a vague warming estimate of >20°C, especially when it follows such a specific lower range of 9°C? Please consider taking them at face value and provide a specific range, or simply use the upper limit of their reconstruction in all instances (e.g. "Miocene warming along the Arctic coast up to 29°C higher than today").

- The authors have addressed this concern with citation of paleo-T estimates from Umbo et al.

- Late Miocene precipitation (800-900 mm MAP) is estimated to be slightly higher than the Pliocene (700-800 mm MAP). As far as I can tell, we aren't provided any indication of modern MAP at the cave site, with which to compare these earlier values. But these data are highly important to assessing the relevance of MAAT changes and the expected ecological evolution of a post-permafrost landscape.

- The authors have addressed this comment with added MAP data.

Line 29-30: "Mean annual ground warming above 0°C..." Is this supposed to read mean-annual ground temperature (MAGT)? If so, please clarify it in the text. Either way, the discussion would benefit from considering the various thresholds for permafrost cover and the relationship between permafrost, ground temperature, and surface air temperature.

- The authors have addressed this comment and now clearly distinguish the role of ground temperature throughout the manuscript.

For example, permafrost-stored soil carbon is already susceptible to mobilization at the continuous-discontinuous boundary (around $-1.7 \pm 0.5^\circ\text{C}$; Obu et al., 2019), which corresponds (in northern Siberia) to MAAT between -9°C and -6°C (Gruber et al., 2012). The modern study site is already within 5°C of this threshold.

Line 31-32: What is the source of this "considerable uncertainty"? Shouldn't sustained MAGT $>0^\circ\text{C}$ do the trick? If so, calculate the difference between present ground temperatures and 0°C , and that's how much warming is needed. Beyond that, it's only a matter of time (which does have some uncertainty, but as far as I can tell, thaw rates are not addressed in this study).

Line 42-52: The site description is good, but it need not be repeated verbatim in the SI and it lacks two vital components: 1) What is the monthly/seasonal precipitation, locally? 2) What are the ground temperatures near the site? Since the focus is on permafrost extent and degradation, ground temperature is a more important component than air.

- The authors have addressed this fully.

U-Pb dating: I congratulate the authors on this stunning geochronological result. Can you clarify in the text, was any attempt made to constrain the range of speleothem-recorded growth? How thin of a snapshot are we talking? I appreciate this may not have been plausible with the sample selection (especially if they're nearly coeval), but if that's the case, there should be mention of how much time feasibly could be recorded by these speleothems.

- The authors have answered this well. The updated text provides a much clearer picture of the chronology and potential time span represented in speleothem collections.

Line 101: "and the fact that any speleothem formation implies a MAAT of $>-5^{\circ}\text{C}$..."

Why -5°C ? Because of conditions around Botovskaya? Either way, this is an oversimplification of cryospheric processes and thresholds. This sentence could be revised to something like:

"Considering today's MAGT of $-XX^{\circ}\text{C}$ at Taba-Ba'astakh and the fact that any speleothem formation implies a MAGT of at least -2°C (i.e. discontinuous permafrost) but more likely $>0^{\circ}\text{C}$..."

- I appreciate the revised discussion in lines 68–73, which shows that paleo-T estimates from 6.6 – 11.1°C preclude the possibility even of discontinuous permafrost, even accounting for uncertainties. However, I feel the authors revert to an oversimplification of these clumped-isotope data, particularly in lines 81–84, where they conflate them with MAAT:

- "Clumped isotope paleotemperatures from Taba-Ba'astakh speleothems show MAAT of $+6.6^{\circ}\text{C}$ - $+11.1^{\circ}\text{C}$, indicating a warming of 18.9°C to 23.4°C above present, considering the difference between the Tortonian MAAT and present day MAAT of -12.3°C ."

- The clumped isotope approach specifically provides an estimate of the fluid temperature during speleothem growth. This temperature is certainly related to, but is not the same as, MAAT at the surface. Drip-water temperature reflects the mean ground and cave-air temperatures at depth, which are insulated by the extent and length of snow cover (among other factors, discussed in my original comments below). High-latitude cave sites, particularly in Siberia, almost invariably exhibit warmer internal temperatures than the mean surface air temperature, some by at least 4°C . It is entirely plausible that a Late Miocene cave temperature of 6.6°C corresponded to a MAAT much closer to 0°C (but certainly less than 6.6°C).

- Calculating a precise Nival offset for the site would be inconsequential to your conclusions herein. It's also impossible to constrain from your dataset. What matters

most is that the cave/ground temperature is likely to be at least $\sim 6.6^{\circ}\text{C}$, which is well above 0°C and thus permafrost-free. Given the importance of this study, I strongly feel it is necessary to avoid such an oversimplification as equating the clumped isotope estimates directly with MAAT.

- I recommend and would appreciate a simple statement in the methods explaining that the authors assume cave temperature and MAAT to be the same and why they think it's reasonable to assume this.

Our reply:

In cold regions, snow indeed acts to insulate the ground, reducing heat loss to the atmosphere (Molnar, 2022). This insulating effect has been shown to lead to cave air temperatures $5\text{-}7^{\circ}\text{C}$ higher than surface air temperatures in cold regions with persistent (ca. 233 days per year) snow cover (Töchterle et al., 2024).

We argue that in our case there would have been minimal nival offset, since the effects of increased ground shading from forest vegetation likely counteracted the insulating effects of winter snow cover. Palynological evidence suggests the Miocene treeline reached 80°N (Steinthorsdottir et al., 2021b), meaning some degree of forest cover at Taba Ba'astakh during that time. Monitoring studies in Eagle Cave, Spain, show reduction of the cave temperature up to 2°C when the overlying vegetation regime shifts from shrub to forest due to changes in insolation and modification of soil properties (Domínguez-Villar et al., 2013). Given the higher latitude of Taba Ba'astakh it is reasonable to assume a reduced impact from insolation shielding compared with Eagle Cave, although a small offset ($< 2^{\circ}\text{C}$) is possible between our cave temperature reconstructions and surface temperature.

The $\text{T}\Delta 47$ reconstructions from Umbo et al. (2025) between 6.6 and 11.1°C suggest a mean annual surface temperature between modern day Stockholm (Moberg, 2021) and London (Met Office, 2016). These places experience 75-100 days of snow in the first case, to 5-10 days of snow in the second case, both substantially less than 233 days of snow cover per year. We therefore suspect limited effect of snow insulation at Taba Bastaakh.

We wrote in the lines 85-92: "The minimum MAAT range that leads to discontinuous permafrost and enables water penetration into the caves is 0°C to -6°C (ref 19), however, clumped isotope analyses performed on four Taba-Ba'astakh speleothems show paleotemperatures (MAGT) of $+6.6^{\circ}\text{C}$ - $+11.1^{\circ}\text{C}$ (ref 20). During the Tortonian some tree growth extended to 80°N (ref 25) and in forested regions with $\text{MAAT} \geq +5^{\circ}\text{C}$, the MAGT are close to MAAT (ref 23). The temperatures between $+6.6^{\circ}\text{C}$ and $+11.1^{\circ}\text{C}$ (ref 20) range between present-day MAAT in Stockholm (ref 21) and London (ref 22), respectively, and it follows that the climate was temperate. If permafrost was absent at this site and further south, this indicates that most of the Siberian landmass and likely similar regions in the Northern Hemisphere were permafrost-free when speleothems formed at Taba-Ba'astakh."

By relating the discussion to MAAT, the complex relationship between ground and air temperatures is missing, which is important in regions like Siberia with high-amplitude seasonality and substantial winter snowpack (though I appreciate that the insulating effect of snow is briefly mentioned):

Smith, M.W., and D.W. Riseborough 1996. Permafrost Monitoring and Detection of Climate Change, *Permafrost and Periglacial Processes*, 7, 301–309.

Wright et al., 2003, Regional-scale permafrost mapping using the TTOP ground temperature model

Gruber et al., 2012 - <https://doi.org/10.5194/tc-6-221-2012>

Obu et al., 2019 - <https://doi.org/10.1016/j.earscirev.2019.04.023>

Peng et al., 2024 - <https://doi.org/10.1088/1748-9326/ad30a5>

The Botovskaya Cave region is not necessarily informative of the air-ground insulating effect along the Arctic coastline, because the determining factors are the relative proportion of freezing and thawing degree days (a function of seasonal amplitude), their respective conductivities, and snow-cover days. Hence that air-ground temperature difference of $\sim 5^{\circ}\text{C}$ near Botovskaya climbs to $\sim 9^{\circ}\text{C}$ near the Taba-Ba'astakh site.

Without specific constraints on changes in winter precipitation and summer-winter amplitude during the Miocene, it is difficult to estimate the magnitude shift in MAAT required to induce permafrost thaw at the modern Arctic coastline (i.e. a $+9^{\circ}\text{C}$ change in MAAT does not necessarily equate to a $+9^{\circ}\text{C}$ change in ground temperature). Please consider incorporating this in the discussion more comprehensively or limit the quantitative estimates to ground-temperature changes as a combined, yet uncertain, function of surface temperature plus winter precipitation. I think the latter almost achieved already but could be improved a little.

- The authors have addressed my concerns about the reference to Botovskaya Cave.

Line 114-118: "Estimates by Schuur et al. suggest that 5-15% of this surficial soil organic carbon would be vulnerable to be released as greenhouse gases within decades in the short-term, while longer-term estimates for soil carbon release are poorly constrained. This means that 30-85 PgC will be released into the atmosphere for the conservative estimate of uniform warming of 9°C across Siberia (Fig. 3-C)."

- The authors have addressed this comment. I appreciate the updated references and discussion.

In my opinion, the application of modeling studies by Schuur et al. needs to be refined, and much of the vast literature since 2015 on this topic has been left out. Consequently, the manuscript oversimplifies cryospheric processes also in terms of carbon cycling.

First, I am not convinced that the cited range of Siberian warming ($9\text{-}29^{\circ}\text{C}$) is realistic for global MAAT forecasts up to $+4.5^{\circ}\text{C}$, unless you can also remove the Greenland ice sheet in a matter of decades this century.

- This comment has been addressed. I think it is now sufficiently communicated that the Miocene analogy is a long-term potential response to high warming scenarios, rather than a near-term expectation.

Second, the estimates for carbon mobilization are based on limited data, hence Schuur et al. caution that several uncertainties exist. These processes have been extensively investigated since Schuur et al. 2015 with updated datasets (refs below), including the find that relatively little of the carbon release during permafrost and peatland warming is from ancient carbon stores, at least during the Early Holocene. Consequently, the manuscript would benefit from refining its discussion to include some of the vast literature post-2015 that has not been cited. Otherwise, it risks oversimplifying cryospheric processes in terms of carbon cycling.

Finally, carbon cycling amid abrupt permafrost degradation is complicated by the specific ecological response, particularly with regard to peatland formation:

Turetsky et al., 2020, <https://www.nature.com/articles/s41561-019-0526-0> (as a start; almost any recent paper by Merritt Turetsky will be highly relevant)
Smith et al., 2004, Siberian peatlands a net carbon sink and global methane source since the Early Holocene
MacDonald et al., 2006, Rapid early development of circumarctic peatlands and atmospheric CH₄ and CO₂ variations.
Alexandrov et al., 2016, The influence of climate on peatland extent in Western Siberia since the Last Glacial Maximum

In short, moisture availability and rates of future warming are important factors in constraining carbon mobilization to atmospheric reservoirs versus terrestrial uptake through peatland expansion. Providing a simple fraction of carbon stored within given isotherms is again misleading, because it belies the complex but well studied dynamics of carbon cycling and storage in permafrost and peatlands. Under what conditions would carbon mobilization to the atmosphere be amplified or mitigated during permafrost thaw in the high Arctic of Siberia?

I hope the authors can develop a more nuanced discussion of these dynamics, because this speleothem-based record will provide an important contribution to the field and to a time and region that is scarcely represented by terrestrial proxy data.

-The authors have addressed this comment in their response and with the updated references.

- Additional comments:

- Line 192-193, Methods: Did you mean to write Mean Ground Annual Temperatures (MGAT) instead of Mean Annual Ground Temperatures (MAGT)? It is now inconsistent with the main text.

This was corrected to “This region is underlain by continuous permafrost with a thickness of 300-500 meters (ref 2) and with MAGT of -9°C - -11°C (ref 2) (Fig. 4).”, line 212-213.

Reviewer #3 (Remarks to the Author):

dear authors,

Thank you for asking my second opinion on the manuscript by Vaks et al. As requested, I have gone over the manuscript and the author’s rebuttal again and I must conclude that the authors did a great job responding to my points. Their explanation of the impact of changes in temperature seasonality on the permafrost is clear, and makes sense to me given the definition of permafrost used by the authors. While I still think that it would be helpful to clarify exactly for which type of future climate scenario the Tortonian is an analogue, I sympathize with the fact that this might be too much detail for the discussion in this manuscript and agree with the authors’ solution in lines 132-134. I also appreciate the improvements to Figures 2 and 3 and the explanation of the age constraints of the speleothem samples. Given the extensive replies of the authors to the other reviewers, **I believe this manuscript is now ready for publication in Nature Communications and would like to congratulate the authors on their nice study.**

Reviewer #4 (Remarks to the Author):

We made other corrections as follows (marked in green):

Lines 22-23: That especially considering that Arctic warming is happening at nearly four times of the global average rate (ref 8).

Line 29: “(mean annual air temperatures (MAAT))”

Line 60-61: “and mean annual ground temperatures (MAGT) of -11°C - -9°C (ref 13) (see also Methods).”

Line 76: “Tortonian stage (11.63 -7.25 Ma)”

Line 77: “sea surface temperatures (SST)”

Line 103: “for Tortonian”

Line 143: “Northern Hemisphere”

Caption of Figure 1, line 179: “Ocean Drilling Program”.

Caption of Figure 2: Title added “Comparison of speleothem deposition period in Taba-Ba’astakh to other climate records.”, line 182; “vertical bold line”, line 184; “light yellow shading”, line 185; “OH-GDGT%-SST from ODP site 910C (purple line and circles) (ref 39)” lines 188-189; and “The light-blue shading on the left shows the period when ice-raftered debris appears in ODP-151-907 borehole (ref 40)” Lines 191-192.

Caption of Figure 3: “soil organic carbon (SOC)” Line 202.

Fig. 3 – in order to meet the needs of colour-blind readers, we removed greenish colours because the soil organic carbon concentrations in present-day permafrost areas are marked in red.

Line 212: “MAGT”

Line 217: “between 05 and 15 August 2014”.

Line 223: “of these exposed speleothems sites”

Line 237: “above the riverbank were STBB-II speleothems were collected”

Lines 254-255: U-Pb chronology of speleothems was performed using the methods, half-lives and age-calculation assumptions described in Vaks et al. (2020) (ref 15) and Mason et al. (2022) (ref 16).

Line 305: “For many less radiogenic samples,...”

Line 335: “land stations were obtained”

Line 339: “A soil organic carbon (SOC)...”

Line 347: “the sum of remaining SOC (in kg/m⁻²)”

The number of figures was reduced to 10 to meet the rules of the journal. This required the following:

The former figures 6, 7 and 8 were combined into the new Figure 6; the former figures 9 and 10 were combined into the new Figure 7; The former figures 11 and 12 were combined into the new Figure 8; The former figures 13 and 14 became 9 and 10 respectively. The references along the manuscript were changed accordingly.

The figure captions were changed accordingly as follows:

Caption of the new Figure 6 (lines 365-372): “Picture of the Taba-Ba’astakh Cliffs with locations of areas on the cliff where speleothems were located, including speleothem sampling site STBB-4 (A, B). Cliffs’ height is 120-140 m, and the speleothem fragments that fell from the cliff were collected on the riverbanks STBB-I and STBB-II, with the ship mooring point in between. Karst caves located on the southern edge of the Taba-

Ba'astakh cliffs (C, D) about 50-60 m below the top of the cliff. The ice filling in caves begins 10-15 m from the entrance (E). One of the rock fragments with a flowstone attached (STBB-II area) is shown in (F). Speleothems on the cliff at the STBB-4 site: Mamillaries (G), Flowstone (H, I), Stalactites (J). (Photos by Osinzev, A. and Vaks, A.)”

Caption of the new Figure 7 (lines 374-382): “Sections of speleothems used in the study with marked growth layers. Flowstone STBB-I-1 (A); Flowstone STBB-4-2 (B); Folia STBB-4-3 (C); Flowstone STBB-4-5 (D); Flowstones STBB-I-2 (E); STBB-I-6 (F); STBB-II-1 (G); STBB-II-5 (H); STBB-II-7 (I); STBB-I-5 (J); STBB-I-12 (K); STBB-II-2 (L); STBB-II-8 (M); and STBB-II-9 (N). Sampling for XRD analysis is marked by red squares. The examples of speleothems’ petrography under the polarizing microscope usually showed the typical columnar crystal structure with no weathering signs, like in flowstone STBB-I-12 (O – cross-polar-light (cpl); P – plain-polar-light (ppl)). In two speleothems, STBB-II-1 and STBB-II-9 along the usual petrography, some weathering signs were found. These include micritization near the surface (STBB-II-1, Q – cpl; R – ppl), or porosity (STBB-II-9, S - cpl, T - ppl). While sampling for chronology, these weathered places were avoided.”

Caption of the new Figure 8 (lines 384-401): “Isochron diagrams showing all Taba-Ba’astakh chronology data. In $^{238}\text{U}/^{206}\text{Pb}$ - $^{208}\text{Pb}/^{206}\text{Pb}$ space (A) all data from samples STBB-II-8, STBB-II-2, STBB-4-2, STBB-4-5, STBB-I-1, STBB-II-5 and STBB-II-7 (selected data in blue) form a near-perfect linear array suggesting that these samples are undisturbed, and have a uniform initial $^{234}\text{U}/^{238}\text{U}$ ratio, uniform age and common Pb composition. Note that the absolute age for these samples cannot be unambiguously determined solely from the ^{238}U - ^{206}Pb system in this instance, because it is not possible to characterise the initial $^{234}\text{U}/^{238}\text{U}$ ratio without additional information. The data from the remaining samples (red) shows considerable excess scatter attributable to open-system behaviour, variation in age, variation in initial $^{234}\text{U}/^{238}\text{U}$ ratio, or variation in common Pb composition, in some combination. In $^{235}\text{U}/^{207}\text{Pb}$ - $^{208}\text{Pb}/^{207}\text{Pb}$ space (B) all but one data ellipses are located on one isochron line. The data from samples STBB-II-8, STBB-II-2, STBB-4-2, STBB-4-5, STBB-I-1, STBB-II-5 and STBB-II-7 are in blue, data from other samples in red. As in $^{238}\text{U}/^{206}\text{Pb}$ - $^{208}\text{Pb}/^{206}\text{Pb}$ isotope space, samples STBB-II-8, STBB-II-2, STBB-4-2, STBB-4-5, STBB-I-1, STBB-II-5 and STBB-II-7 form a near-perfect linear array, again suggesting a common $^{235}\text{U}/^{207}\text{Pb}$ age of c. 8.8 Ma for these samples. Data from other samples are also mostly consistent with the same isochron and show less scatter in $^{235}\text{U}/^{207}\text{Pb}$ - $^{208}\text{Pb}/^{207}\text{Pb}$ isotope space than $^{238}\text{U}/^{206}\text{Pb}$ - $^{208}\text{Pb}/^{206}\text{Pb}$ isotope space suggesting that some of the scatter in the latter is probably attributable to variation in the initial $^{234}\text{U}/^{238}\text{U}$ ratio, which does not affect the ^{235}U - ^{207}Pb system. A common $^{208}\text{Pb}/^{207}\text{Pb}$ value of 2.45 ± 0.15 is estimated based on the isochron fit to samples STBB-II-8, STBB-II-2, STBB-4-2, STBB-4-5, STBB-I-1, STBB-II-5 and STBB-II-7.”

Caption of the new Figure 9 (lines 404-407): “Analytical uncertainty as function of non-radiogenic ^{207}Pb in the samples. The isochron $^{208}\text{Pb}/^{207}\text{Pb}$ - $^{235}\text{U}/^{207}\text{Pb}$ age of 8.68 ± 0.09 Ma is marked by the grey line. The samples become less radiogenic from left to right, causing the $^{208}\text{Pb}/^{207}\text{Pb}$ ratio to increase from 0 (left) to 2.5 (right). The age uncertainty (vertical error bars, 2σ) becomes larger in this direction as well.”

Caption of the new Figure 10: “Reduction of the permafrost area⁴³ in the Northern Hemisphere as a function of an increase in MAAT¹¹ at a 2°C interval. 20°C+ warming is likely to result in the complete disappearance of permafrost.”

We also added the following sections:

Lines 414-416: “Data availability: Source data (Datasets 1 and 2) are provided with this paper as supplementary information and can be accessed via zenodo.org, doi: **10.5281/zenodo.15194760.**”

Lines 447-448: “Competing interests: Authors have no competing interests.”

Bibliography: we added the reference of “Ершов, Е. Д. (Геокриологическая карта СССР, масштаб 1 : 2.5 млн , Мин. геологии СССР и МГУ, 1991).”, the original reference of the Soviet map the Fig. 4 is based on.

We changed the corresponding author to Sebastian F.M. Breitenbach (sebastian.breitenbach@northumbria.ac.uk) in order to assist the University of Northumbria library to pay the manuscript costs.